# Sam68 splicing regulation contributes to motor unit establishment in the postnatal skeletal muscle

Elisa De Paola[1,2,*], Laura Forcina[3,*], Laura Pelosi[3] , Simona Pisu[3], Piergiorgio La Rosa[2] , Eleonora Cesari[2,4], Carmine Nicoletti[3], Luca Madaro[4], Neri Mercatelli[1,2], Filippo Biamonte[5], Annalisa Nobili[2,6], Marcello D'Amelio[2,6], Marco De Bardi[2] , Elisabetta Volpe[2], Daniela Caporossi[1], Claudio Sette[2,4,†], Antonio Musarò[3,†], Maria Paola Paronetto[1,2,†]

RNA-binding proteins orchestrate the composite life of RNA molecules and impact most physiological processes, thus underlying complex phenotypes. The RNA-binding protein Sam68 regulates differentiation processes by modulating splicing, polyadenylation, and stability of select transcripts. Herein, we found that $Sam68^{-/-}$ mice display altered regulation of alternative splicing in the spinal cord of key target genes involved in synaptic functions. Analysis of the motor units revealed that Sam68 ablation impairs the establishment of neuromuscular junctions and causes progressive loss of motor neurons in the spinal cord. Importantly, alterations of neuromuscular junction morphology and properties in $Sam68^{-/-}$ mice correlate with defects in muscle and motor unit integrity. $Sam68^{-/-}$ muscles display defects in postnatal development, with manifest signs of atrophy. Furthermore, fast-twitch muscles in $Sam68^{-/-}$ mice show structural features typical of slow-twitch muscles, suggesting alterations in the metabolic and functional properties of myofibers. Collectively, our data identify a key role for Sam68 in muscle development and suggest that proper establishment of motor units requires timely expression of synaptic splice variants.

## Introduction

Execution of gene expression programs in eukaryotic cells requires a complex network of regulative processes that integrate nuclear transcription and processing of the pre-mRNA with cytosolic utilization of the mature transcripts. In this regulative network, a crucial role is played by RNA-binding protein (RBPs), which associate with transcripts during their whole life cycle and determine, in time and space, the availability of specific transcript variants in the cell (Gerstberger et al, 2014; Jangi & Sharp, 2014). A key step regulated by many RBPs is the processing of the nascent transcripts, including selective assortment of exons through alternative splicing (Black, 2003) and alternative termination and polyadenylation (Tian & Manley, 2017). These highly flexible and tunable processes respond to internal and external cues and allow production of multiple transcripts from each gene (Barbosa-Morais et al, 2012; Irimia & Blencowe, 2012). Because splice variants often display different activities and/or patterns of expression, alternative splicing contributes to amplification of the coding potential of the genome and allows expression of the appropriate proteome repertoire required to execute specialized cell functions (Fu & Ares, 2014; Paronetto et al, 2016).

RBPs can determine tissue-specific splicing patterns through recognition of splicing enhancer and silencer elements in the pre-mRNA, consequent modulation of the assembly of the spliceosome machinery and selection of tissue-specific exon usage (Pan et al, 2004; Kalsotra & Cooper, 2011). Protooncogene SRC, Rous sarcoma (SRC) associated in mitosis of 68 kD (Sam68) belongs to the STAR (Signal Transduction and Activation of RNA metabolism) family of RBPs, which regulate several aspects of RNA metabolism (Vernet & Artzt, 1997; Lukong & Richard, 2003; Frisone et al, 2015). STAR proteins are characterized by a highly conserved RNA-binding domain comprising a central human heterogeneous nuclear ribonucleoprotein (hnRNP) K homology (KH) domain flanked by two homologous regions, termed Qua1 and Qua2 and regulatory regions outside of the RNA-binding domain (Vernet & Artzt, 1997). In particular, Sam68 is subjected to several posttranslational modifications that modulate its subcellular localization, interaction with signaling proteins, and affinity for target RNAs (Lukong & Richard, 2003; Paronetto et al, 2003; Sette, 2010; Frisone et al, 2015).

---

[1]Department of Movement, Human and Health Sciences, University of Rome "Foro Italico," Rome, Italy   [2]IRCCS (Institute for Treatment and Research) Fondazione Santa Lucia, Rome, Italy   [3]Laboratory Affiliated to Istituto Pasteur–Fondazione Cenci Bolognetti, DAHFMO–Unit of Histology and Medical Embryology, Sapienza University of Rome, Rome, Italy   [4]Institute of Human Anatomy and Cell Biology, Catholic University of the Sacred Heart, Rome, Italy   [5]Institute of Biochemistry and Clinical Biochemistry, Fondazione Policlinico Universitario A. Gemelli IRCCS, Rome, Italy   [6]Department of Medicine, University Campus-Biomedico, Rome, Italy

Correspondence: mariapaola.paronetto@uniroma4.it; antonio.musaro@uniroma1.it; claudio.sette@unicatt.it
*Elisa De Paola and Laura Forcina contributed equally to this work
†Claudio Sette, Antonio Musarò, and Maria Paola Paronetto contributed equally to this work

---

Elucidation of the physiological roles of Sam68 has been facilitated by the generation of a knockout mouse model. Whereas $Sam68^{-/-}$ mice display significant (~30%) perinatal lethality, surviving animals reach adulthood and can be investigated (Richard et al, 2005). MEF deficient of Sam68 are impaired in adipocyte differentiation (Richard et al, 2005; Huot et al, 2012), suggesting a role for this RBP in the regulation of the balance between adipogenic and osteogenic differentiation. Accordingly, $Sam68^{-/-}$ mice are protected from age-induced osteoporosis and display preserved bone density (Richard et al, 2005). Moreover, $Sam68^{-/-}$ male mice are infertile (Paronetto et al, 2009), whereas females display delayed mammary gland development and reduced fertility (Richard et al, 2008; Bianchi et al, 2010). Sam68 deficiency was also reported to impair motor coordination (Lukong & Richard, 2008) and social behavior (Farini et al, 2020). On the other hand, Sam68 has been involved in the pathogenesis of fragile X-associated tremor/ataxia syndrome (Sellier et al, 2010) and spinal muscular atrophy (Pedrotti et al, 2010; Pagliarini et al, 2015), as well as in brain development and function (Iijima et al, 2011; Danilenko et al, 2017; Witte et al, 2019; Farini et al, 2020) through modulation of neuron-specific splicing events.

In this study, we found that ablation of Sam68 affects the neuromuscular strength and causes loss of motor neurons in the first month of age. Importantly, these morphological and functional defects were associated with defective splicing of several genes involved in pre- and post-synaptic functions in the spinal cord, indicating the requirement of Sam68 for proper establishment of neuromuscular junctions (NMJs) in postnatal mice. We also describe that $Sam68^{-/-}$ mice display reduced muscle mass compared with wild-type littermates, with fibers being characterized by a smaller cross-sectional area (CSA). This phenotype is particularly strong at 12 wk of age, indicating progressive defects in the postnatal muscle development. $Sam68^{-/-}$ muscles also show a switch from fast-twitch to slow-twitch fibers and manifest signs of atrophy, suggesting alterations in the metabolic activity and functional properties of muscle fibers. These findings identify a key role for Sam68 in muscle development and suggest that proper establishment of motor neuron connections with muscle fibers requires timely expression of splice variants involved in synapse composition and function.

## Results

### Sam68 regulates splicing of synaptic genes in the spinal cord

Proper muscle innervation requires establishment of synaptic connection between motor neurons and both afferent fibers and effector muscle fibers. Previous work indicated that Sam68 is highly expressed in the motor neurons of the spinal cord (Pagliarini et al, 2015), suggesting an important function of this protein in these cells. Sam68 is known to modulate splicing of several genes encoding for synaptic proteins (Iijima et al, 2011; Danilenko et al, 2017; Witte et al, 2019; Farini et al, 2020). To test whether Sam68 regulates splicing of such synaptic genes in the spinal cord, we selected splicing events from several RNA-sequencing and

microarray experiments carried out in Sam68-depleted mouse tissues or cells (Ehrmann et al, 2008; Chawla et al, 2009; Iijima et al, 2011; Paronetto et al, 2011; La Rosa et al, 2016; Danilenko et al, 2017; Witte et al, 2019; Farini et al, 2020). Spinal cord tissue was isolated from wild-type and $Sam68^{-/-}$ mice and the differential expression of Sam68 was confirmed in both spinal cord and muscle tissues (Fig S1A and B). We found that deficiency of Sam68 favors the inclusion of exon 20 (AS4) of *Nrxn1* in the spinal cord (Figs 1A and S2A), a splicing event that determines the affinity of neurexin proteins for synaptic receptors (Südhof, 2017) and directly affects synapse function (Aoto et al, 2013; Traunmüller et al, 2016). By contrast, the homologous *Nrxn2* AS4 exon, which is a specific target of SLM2 but not Sam68 (Danilenko et al, 2017), was unaffected (Fig S2B). Sam68 ablation also affected the alternative inclusion of exon 22 of the *syntaxin-binding protein 5-like* (*Stxbp5l*, alternatively known as *tomosyn-2*) gene (Fig 1B), encoding a protein required for normal motor performance and involved in neurotransmission at motor endplates (Geerts et al, 2015). Furthermore, we investigated the regulation of postsynaptic scaffolding molecules that are potential targets of Sam68 and play a central role in synaptic functions or glutamatergic synapses (Witte et al, 2019; Farini et al, 2020). Among them, we selected either of the GABAergic post-synaptic proteins collybistin (*Arhgef9*), gephyrin (*Gphn*), and densin-180 (*Lrrc7*), the subunits of the glutaminergic AMPA receptor (AMPAR) *Gria2* and *Gria3*, which mediates the vast majority of fast synaptic transmission in the central nervous system (Li et al, 2016), and of the ATP-sensitive potassium channel SUR2, encoded by the *Abcc9* gene (Nichols, 2016). Importantly, splicing of all these genes was significantly dysregulated in $Sam68^{-/-}$ spinal cord (Fig 1C–I). In particular, inclusion of exon 10a in *Arhgef9* (Fig 1C), of exon 10 in *Gphn* (Fig 1D), and of exon 24 in *Lrrc7* (Fig 1E) was significantly increased in $Sam68^{-/-}$ mice. Furthermore, RT-qPCR analysis revealed an increase in the retention of intron 11 of *Gria2*, intron 12 of *Gria3*, intron 16 of *Ncam2*, and of intron 5 of *Abcc9* transcripts in $Sam68^{-/-}$ mice compared with wild-type littermates (Fig 1F–I), events that are likely associated with premature termination of the transcripts and reduction of the functional proteins (Farini et al, 2020).

One of the best characterized splicing events regulated by Sam68 in mouse tissues is the repression of exon 8 inclusion in the *ε* sarcoglycan (*Sgce*) mRNA (Chawla et al, 2009; Paronetto et al, 2011). Accordingly, skipping of exon 8 was completely abolished in $Sam68^{-/-}$ spinal cord (Fig S2C). Notably, mutations in the human *SGCE* gene cause the movement disorder myoclonus dystonia (Zimprich et al, 2001) and autosomal recessive limb-girdle muscular dystrophies (Lim & Campbell, 1998). Furthermore, lack of Sam68 was shown to favor splicing of an inactive truncated variant of aldehyde dehydrogenase 1A3 (*Aldh1A3*) (La Rosa et al, 2016), a metabolic enzyme which promotes the release of energy through anaerobic glycolysis (Mao et al, 2013) and, therefore, relevant for muscle performance. This alternative splice variant of *Aldh1A3* was upregulated in $Sam68^{-/-}$ spinal cord (Fig S2D), further confirming the widespread impact of Sam68 expression on splicing regulation in this tissue.

To evaluate if the splicing defects detected in the spinal cord of $Sam68^{-/-}$ mice were intrinsic to the motor neurons, we isolated them by laser microdissection from wild-type and knockout tissues (Fig S3A). Importantly, splicing analysis of selected genes (*Nrxn1,*

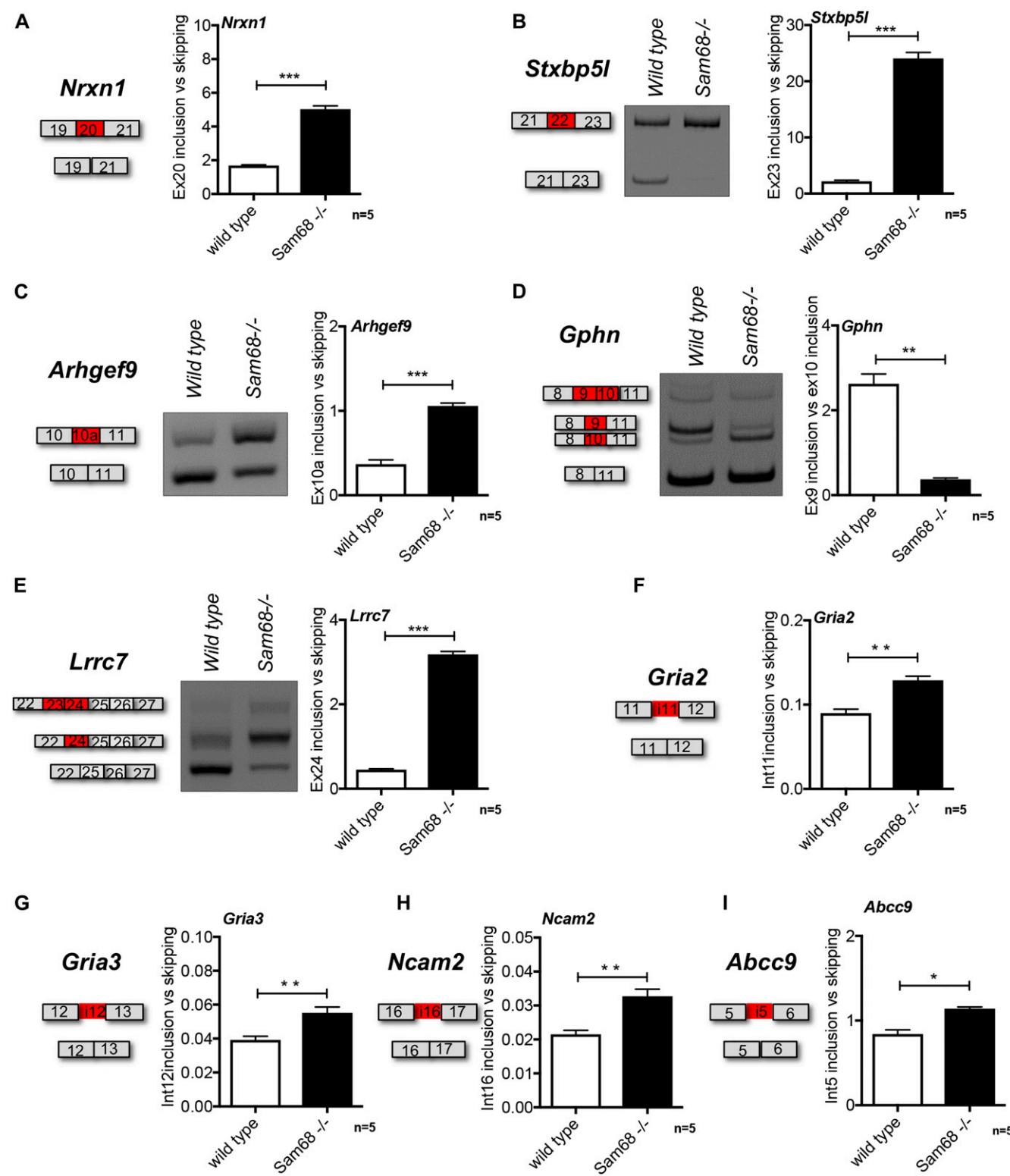

**Figure 1. Sam68 regulates a set of alternatively spliced events in the spinal cord.**
**(A, B, C, D, E, F, G, H, I)** Scheme of the regulated event, representative gel images and densitometric analysis of RT-qPCR (*Nrxn1*, *Gria2*, *Gria3*, *Ncam2*, and *Abcc9*) and RT-PCR (*Stxbp5l*, *Lrrc7*, *Gphn*, and *Arhgef9*) analyses of alternative splicing events differentially regulated between the spinal cord of 45-dpp *Sam68*$^{-/-}$ and control mice. The graph shows the densitometric analysis of the ratio between included and skipped exons (n = 5; mean ± SEM), or retained introns versus skipped. For each experiment, DNAse digestion have been performed. Statistical analysis was performed by *t* test (*$P < 0.05$, **$P < 0.01$, ***$P < 0.001$).

*Arhgef9*, *Stxbp5l*, *Gria2*, and *Gria3*) confirmed the regulation of the Sam68-dependent events also in the isolated motor neurons (Fig S3B–F). Collectively, these data indicate a key role for Sam68 in the regulation of alternative splicing of synaptic genes in the motor neurons of the spinal cord.

### Sam68 deficiency affects integrity of NMJs and motor neuron viability

Next, we asked whether impairment of this splicing program may affect NMJ assembly and motor neuron integrity. Motor endplates in muscles were analyzed by confocal microscopy to detect bungarotoxin, which specifically stains the subunit of acetylcholine receptors (AChRs), synaptophysin, a glycoprotein present in presynaptic vesicles that marks nerve terminals, and neurofilament subunits as a general axonal marker (Fig S4A). As shown in Fig 2A, *Sam68*$^{-/-}$ mice displayed a significant increase in the number of NMJs (Fig 2B), which was accompanied by a reduction in their size (Fig 2C) with a significantly smaller median frequency distribution of the NMJ area (Fig 2D). To further characterize this phenotype, we measured the cellular co-localization between synaptophysin and bungarotoxin signals. In *Sam68*$^{-/-}$ mice, we observed a significant decrease in the overlaid signals, along with an increase in the non-overlaid signals (Fig S4A and B) demonstrating the presence of altered junctions.

Defects in the NMJs can impinge on motor neuron viability by a dying back process, as in amyotrophic lateral sclerosis (Dadon-Nachum et al, 2011). To test whether lack of Sam68 affected the number of motor neurons, coronal sections of the lumbar spinal cord (L1–L5) region were stained with Nissl substance (Fig 2E and F), which intensively marks ChAT-positive motor neurons (Fig S5). Remarkably, *Sam68*$^{-/-}$ mice showed a progressive loss of motor neurons, which was barely detected at 1 wk of age and reached a significant reduction (~50%) by 4 wk of age (Fig 2G). Furthermore, these findings were corroborated by analysis of the sciatic nerve, which indicated clear signs of degeneration (Fig 2H) due to a striking increase in degenerated neuronal axons (Fig 2I). Last, such dramatic loss of motor neurons significantly impacted on the neuromuscular strength of *Sam68*$^{-/-}$ mice, as indicated by their inability to remain on the wire lid for more than 30 s in the *Hanging-wire* test (Oliván et al, 2015), with respect to the more than 150 s observed with wild-type littermates (Fig 2J). These data strongly suggest that ablation of Sam68 function causes aberrant innervation of muscle fibers and loss of motor neurons in the spinal cord.

### Sam68$^{-/-}$ muscles display defects in postnatal muscle development

Muscle denervation is known to cause atrophy and loss of fast-glycolytic fibers (Niederle & Mayr, 1978; Rocchi et al, 2016). To investigate whether aberrant innervation resulted in an atrophic phenotype of *Sam68*$^{-/-}$ muscles, we first monitored body weight in the first 3 mo of age. *Sam68*$^{-/-}$ mice showed reduced weight already at 1 wk of age, and this phenotype became more evident at 4 wk of age (Fig 3A and B). To examine whether muscle mass was specifically reduced, we weighted gastrocnemius (GA), soleus (Sol), tibialis anterior (TA), and extensor digitorum longus (EDL), at 4 and 12 wk of age. Although all muscles examined were smaller at both

ages, normalization for the total body weight indicated that the specific reduction in muscle mass was significant only at 12 wk of age (Fig 3C). These results suggest that Sam68 deficiency mainly impairs muscle mass gain occurring between 4 and 12 wk of age.

Histological analysis of the *Sam68*$^{-/-}$ TA muscle did not reveal overt myopathy, although a reduction in muscle fiber size was evident (Fig 4A). Morphometric quantitative analysis confirmed that *Sam68*$^{-/-}$ muscle fibers displayed a significant reduction (24%) in the CSA already at 4 wk of age (Fig 4B). This defect was strongly enhanced in 12-wk-old *Sam68*$^{-/-}$ mice (Fig 4B), where the CSA of *Sam68*$^{-/-}$ fibers was ~50% with respect to their wild-type counterparts (Fig 4B). At 4 wk of age, the median CSA of wild-type muscle fibers was 996 $\mu m^2$, whereas it was reduced to 720 $\mu m^2$ in *Sam68*$^{-/-}$ muscle fibers (Fig 4C). This difference was exacerbated at 12 wk of age, when the median CSA was 2,651 $\mu m^2$ in the wild-type and 1,180 $\mu m^2$ in *Sam68*$^{-/-}$ muscle (Fig 4C). Thus, impaired development of muscle fibers correlated with the diminished growth of *Sam68*$^{-/-}$ muscles specifically between 4 and 12 wk of age (Fig 3C). Because the number of fibers was not impaired in *Sam68*$^{-/-}$ muscles at both ages (Fig 4D), these observations indicate that lack of Sam68 specifically affects fiber growth.

### Sam68 ablation promotes atrophy in skeletal muscle

To define whether the reduced size of the *Sam68*$^{-/-}$ fibers is accompanied by features of atrophy (Bonaldo & Sandri, 2013), we analyzed the expression levels of two atrophy-related genes: the ubiquitin ligases *Fbxo32* (MAFbx/atrogin-1) and *Trim63* (muscle RING-finger 1, MuRF1). RT-qPCR analyses showed a significant up-regulation (approximately two to threefold) of both atrophy markers in *Sam68*$^{-/-}$ muscles with respect to age-matched wild-type mice (Fig 5A and B). Importantly, such increase in transcript levels was accompanied by up-regulation of the corresponding atrogin-1 and MuRF1 proteins (Fig 5C). Likewise, expression of the FoxO3A transcription factor, which plays a key role in the regulation of muscle atrophy in vivo by activating transcription of both atrogin-1 and MuRF1 (Sandri et al, 2004; Senf et al, 2010), was also increased more than twofold in *Sam68*$^{-/-}$ muscles (Fig 5D). On the other hand, miR-23a, a negative regulator of atrogin-1 and *MuRF1* transcripts (Wada et al, 2011), was down-regulated (Fig 5E). These results suggest that an atrophic program is induced in *Sam68*$^{-/-}$ muscles and likely contributes to the reduced growth of muscle fibers.

Muscle atrophy can be induced by defective innervation. In line with this notion, *Sam68*$^{-/-}$ muscles also exhibited a significant increase in the expression of *myogenin* (Figs 5F and S6), a transcription factor required for the expression of atrogin-1 and MuRF-1 during denervation-induced atrophy (Moresi et al, 2010). Furthermore, flow cytometry analysis of cells isolated from the *Sam68*$^{-/-}$ hind limb indicated a higher percentage of fibro-adipogenic progenitors (FAPs) (Fig 5G). Importantly, FAPs represent precursor cells that accumulate in the muscle in response to loss of NMJ integrity during acute and progressive denervation, such as spinal cord injury or neurodegeneration-associated atrophy (spinal muscular atrophy and amyotrophic lateral sclerosis) (Madaro et al, 2018). These observations suggest that impaired establishment of NMJs in *Sam68*$^{-/-}$ mice leads to motor neuron death, muscle atrophy, and reduced muscle growth in the first months of postnatal life.

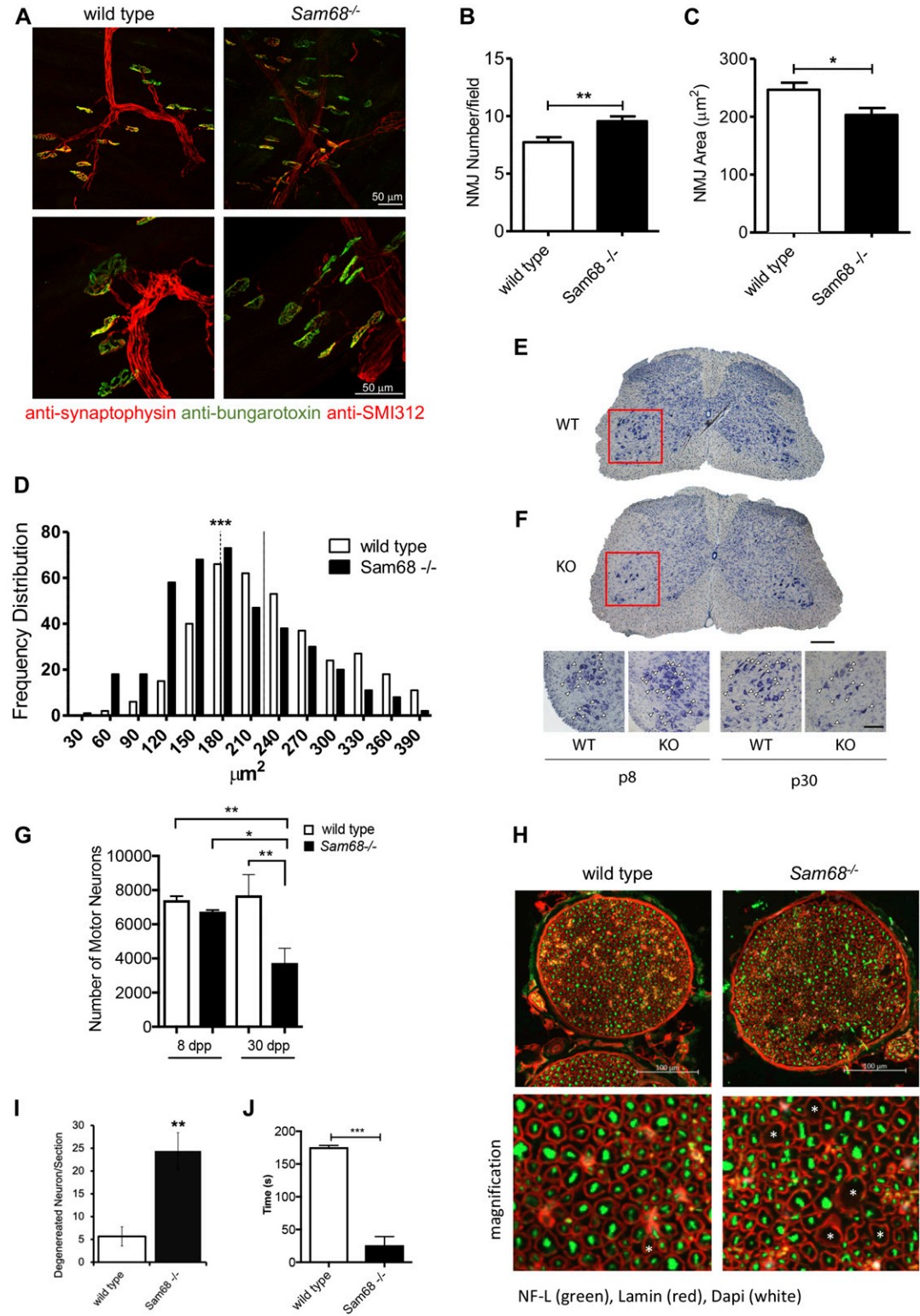

**Figure 2. Sam68 deficiency affects motor neurons upon postnatal muscle development.**
**(A)** NMJ immunofluorescence images from tibialis anterior (TA) of 30-dpp *Sam68*[−/−] and control mice. Postsynaptic (AChRs stained with α-bungarotoxin; green), presynaptic (stained with α-synaptophysin; red), and motor neuron axons (stained with an antibody against the heavy chain of neurofilament; red) were visualized. Scale bars: 50 μm; magnification: 40× (upper panels) and 80× (lower panels). **(B)** Bar graph showing the number of NMJs from TA of 30-dpp *Sam68*[−/−] and control mice (n = 3; mean ± SEM). *P*-value was determined by *t* test (**$P < 0.01$). **(C)** Bar graph showing the area of NMJs from TA of 30-dpp *Sam68*[−/−] and control mice (n = 3; mean ± SEM). *P*-value was determined by *t* test (*$P < 0.05$). **(D)** Frequency distribution of the NMJs areas from TA of 30-dpp *Sam68*[−/−] and control mice. The colored lines and dotted line (*Sam68*[−/−] mice)

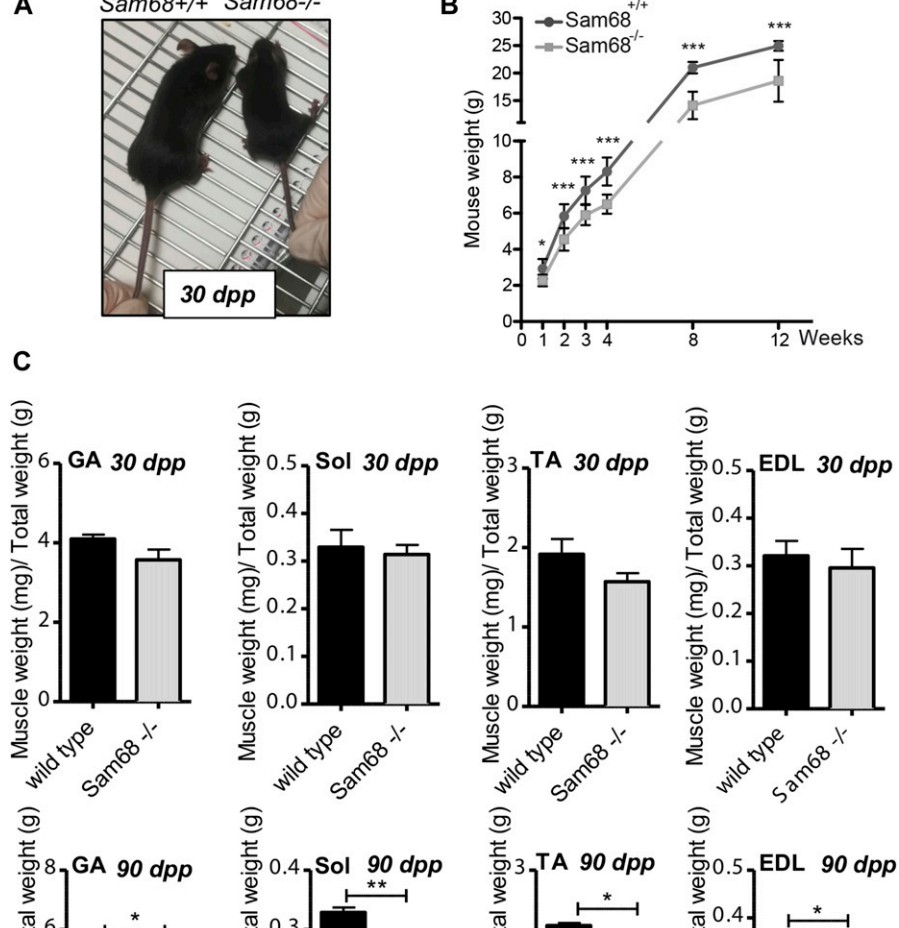

**Figure 3. Sam68 deficiency affects muscle size and mass.**

**(A)** Pictures of $Sam68^{-/-}$ and wild-type littermates at 30 dpp. **(B)** Growth curve of $Sam68^{-/-}$ and control mice. Mice of each genotype were weighed every week. Individual data points represent means ± SEM (n = 22 for $Sam68^{-/-}$; n = 38 for controls); $P$-value was determined by $t$ test (*$P < 0.05$, ***$P < 0.001$). **(C)** Weight was measured of different muscle and normalized to total weight in 30- and 90-d-old mice. Individual data points represent means ± SEM (n = 6); $P$-value was determined by $t$ test (*$P < 0.05$, ***$P < 0.01$).

## Sam68$^{-/-}$ muscles display alteration in fiber type composition

Muscle atrophy associated with motor neuron degeneration is often paralleled by an alteration in fiber type composition (Frey et al, 2000; Hegedus et al, 2008; Rocchi et al, 2016). In adult skeletal muscle, slow (type I)- and fast (type IIA, IIB, and IIX)-twitch myofibers are characterized by different myosin isoforms (Schiaffino & Reggiani, 2011). Analysis by RT-qPCR revealed a dramatic increase in the expression of the slow-twitch type I myosin heavy chain (*MhcI*) isoform in $Sam68^{-/-}$ muscles, whereas expression of fast-twitch specific isoforms (*MhcIIa*, *Mhc2b*, and *Mhc2x*) was not significantly affected (Fig 6A). Accordingly, immunofluorescence analysis

---

represent the median value for each genotype: wild type (n = 3), median size = 223 $\mu m^2$, $Sam68^{-/-}$ (n = 3) median size = 181 $\mu m^2$; statistical analysis of the median values was performed by one-way ANOVA test. **(E)** Representative image of a Nissl-stained spinal cord section of the lumbar spinal cord from 30-dpp WT and KO mice. The red square highlights the ventral horn of the spinal cord analyzed. Bar, 200 $\mu m$. **(F)** Higher magnification of sections of the ventral lumbar spinal cord from 8- to 30-dpp WT and Sam68 KO mice. Motor neurons are indicated by arrowheads. Bar, 100 $\mu m$. **(F, G)** Bar graph representing motor neuron counts (mean ± SEM; n = 3) in lumbar spinal cord from mice described in (F). Statistical analysis was performed by two-way ANOVA test followed by Bonferroni's multiple comparison post-test (*$P < 0.05$, **$P < 0.01$). **(H)** Representative images of sciatic nerve from 45 dpp WT or KO mice, stained with neurofilament (in green) and Laminin (in red); DAPI was used for the staining of the nuclei. **(I)** Bar graphs representing the number of degenerated neurons per sections from three independent experiments. $P$-value was determined by $t$ test (**$P < 0.01$). **(J)** Bar graphs represent time (s) in *hanging-wire test* of 45-dpp wild-type and $Sam68^{-/-}$ mice; (WT n = 7, $Sam68^{-/-}$ n = 7). Data are shown as mean ± SEM. $P$-value was determined by $t$ test (***$P < 0.001$).

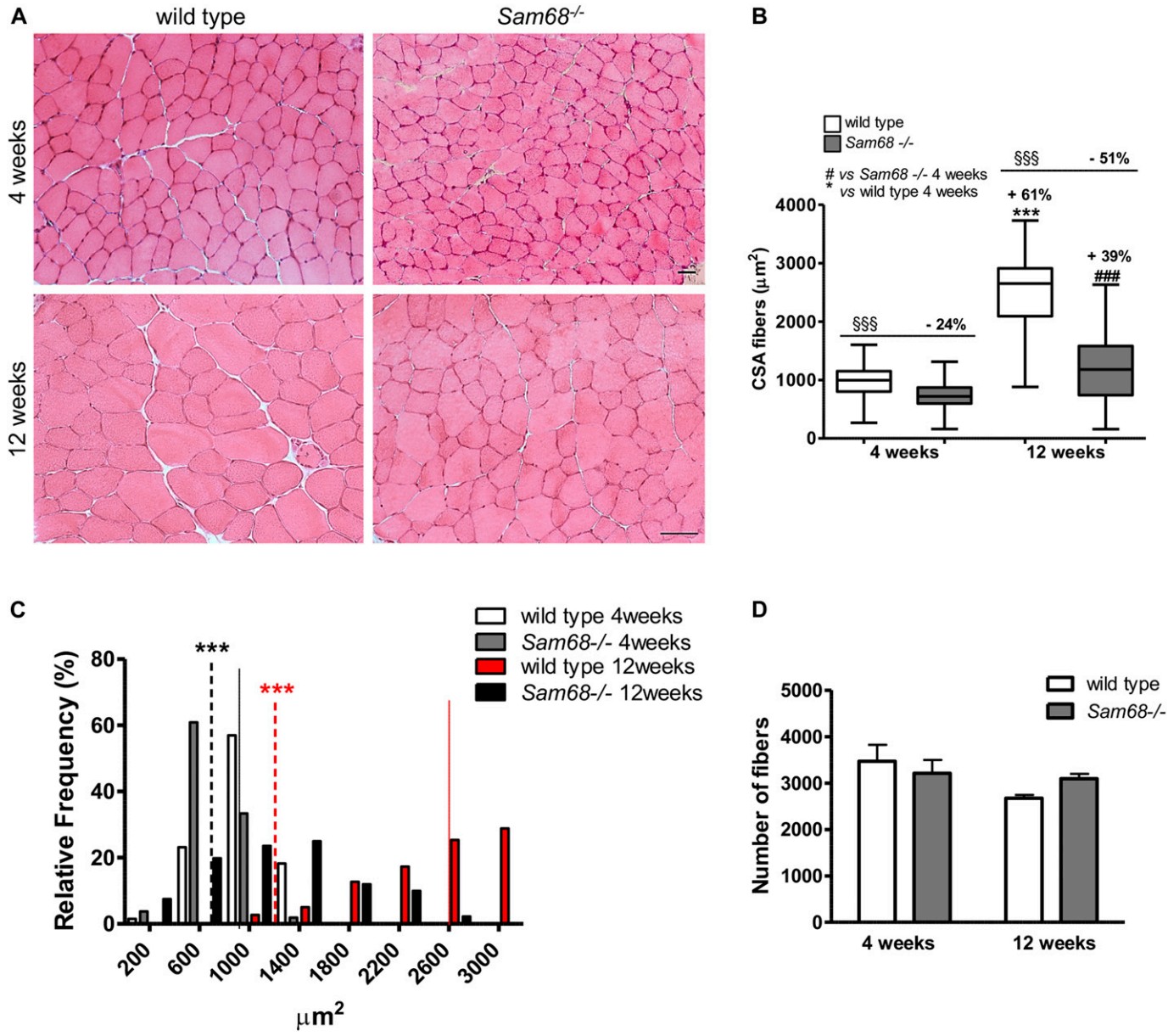

**Figure 4. Sam68$^{-/-}$ muscles display defects in postnatal muscle development.**
**(A)** Hematoxylin and eosin (H&E) staining of muscle cross sections from the tibialis anterior (TA) muscle of 30- and 90-d-old mice and control mice; scale bar: 50 μm; magnification: 20×. **(B)** Quantification of the cross-sectional area of muscle fibers in TA muscle of 30- and 90-d-old Sam68$^{-/-}$ and control mice. Individual data points represent means ± SEM (n = 3); one-way ANOVA (***$P < 0.001$ versus wild-type 4 wk; ###$P < 0.001$ versus Sam68$^{-/-}$ 4 wk; §§§$P < 0.001$ versus relative wild-type control). **(C)** Frequency distribution of muscle fibers areas from TA of 4 and 12 wk Sam68$^{-/-}$ and control mice. The colored lines (wild type) and dotted line (Sam68$^{-/-}$ mice) represent the median value for each genotype: wild-type 4 wk (n = 3) median size = 996 μm$^2$, Sam68$^{-/-}$ 4 wk (n = 3) median size = 720 μm$^2$, wild-type 12 wk (n = 3) median size = 2,651 μm$^2$, Sam68$^{-/-}$ 12 wk (n = 3) median size = 1,180 μm$^2$; individual data points represent means ± SEM (n = 3); one-way ANOVA (***$P < 0.001$). **(D)** Quantification of the number of muscle fibers in TA muscle of 30- and 90-d-old mice and control mice. Individual data points represent means ± SEM (n = 3).

confirmed a significant increase in MHC type I–expressing fibers in Sam68$^{-/-}$ muscles with respect to wild-type littermates (Fig 6B and C). These experiments indicate that atrophic phenotype of Sam68$^{-/-}$ muscles is associated with a fiber type–specific dysregulation.

## Sam68 deficiency affects muscle performance

To determine whether the defects observed in Sam68$^{-/-}$ muscles affect their capacity to produce force, we compared mechanical

parameters of the soleus and EDL muscles of 3-mo-old wild-type and Sam68$^{-/-}$ mice. EDL is a typical "fast" muscle containing 94.2% of fast-twitch fibers (Musarò et al, 2001; Del Prete et al, 2008), whereas soleus is a typical "slow" muscle formed by 60% of slow-twitch fibers (Asmussen et al, 2003; Del Prete et al, 2008). Both EDL and soleus from Sam68$^{-/-}$ mice showed alterations in tetanic force (Fmax) (Fig 7A). However, this difference was essentially due to the reduction in muscle mass. Indeed, Sam68$^{-/-}$ EDL and soleus muscles showed significant reduction in CSA (Fig S7A), like the TA,

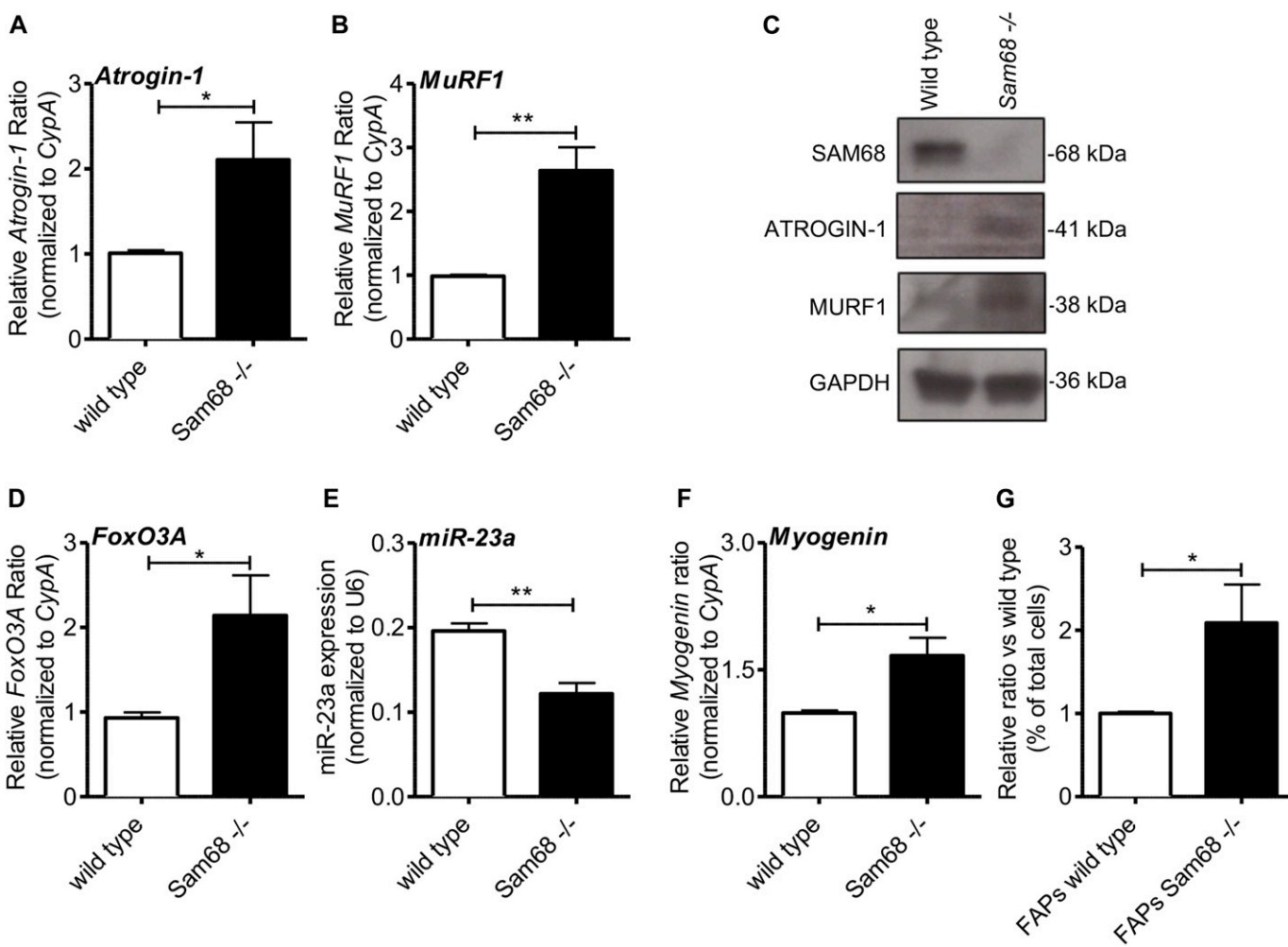

**Figure 5. Sam68 deficiency induces up-regulation of atrogin-1 and MuRF1.**
**(A)** RT-qPCR analysis showing the levels of *atrogin-1* and *MuRF1* transcripts in tibialis anterior (TA) of 30-d-old mice normalized for the levels of the housekeeping gene *CypA* (n = 5; means ± SEM); *P*-value was determined by *t* test (\**P* < 0.05, \*\**P* < 0.01). **(B)** RT-qPCR analysis of miR-23a expression levels in TA of 30-d-old mice normalized for the levels of the housekeeping gene *U6* (n = 5; means ± SEM); *P*-value was determined by *t* test (\*\**P* < 0.01). **(C)** Western blot analysis for SAM68, ATROGIN-1, and MURF1 proteins using TA muscle of 30-d-old mice. GAPDH is shown as a loading control. **(D)** RT-qPCR analysis of *FoXO3A* mRNA levels in TA of 30-d-old mice normalized for the levels of the housekeeping gene *CypA* (n = 8; means ± SEM); *P*-value was determined by *t* test (\**P* < 0.05). **(E)** RT-qPCR analysis of miR-23a expression levels in TA of 30-d-old mice normalized for the levels of the housekeeping gene *U6* (n = 5; means ± SEM); *P*-value was determined by *t* test (\*\**P* < 0.01). **(F)** RT-qPCR analysis of *myogenin* mRNA levels in TA of 30-d-old mice normalized for the levels of the housekeeping gene *CypA* (n = 8). Individual data points represent means ± SEM; *P*-value was determined by *t* test (\**P* < 0.05). **(G)** Bar graph shows the percentage of fibro-adipogenic progenitors, analyzed by flow cytometry, from the whole hind limbs muscles of 60-dpp *Sam68*$^{-/-}$ and control mice (mean ± SEM; n = 4); *P*-value was determined by *t* test (\**P* < 0.05).

and normalization of the *Fmax* for muscle size abolished the difference observed with wild-type muscles (Fig 7B).

Next, we evaluated the maximum value of force derivative to measure the speed of force production (dF/dT). Notably, *Sam68*$^{-/-}$ EDL and soleus muscles had a lower rate of force generation than wild-type muscles (Fig 7C). Because power is the product between force and velocity ($F*v_{max}$), we also evaluated this isotonic parameter. Indeed, we observed a decrease in muscular power both in EDL and soleus muscles of *Sam68*$^{-/-}$ mice, mainly because of the decrease in muscle mass and in the speed of force production (Fig 7D). Analysis of MHC isoform expression by RT-qPCR revealed an increase in the expression of MHC type I also in the EDL and in the soleus (Fig S7B and C), as observed in the TA muscle (Fig 6A). Because the lower speed of force production is a feature of slower

muscles, these data highlight alterations in the mechanical properties of *Sam68*$^{-/-}$ muscle.

## Discussion

In this study, we have identified Sam68 as a new physiological regulator of the muscle motor unit. In particular, our findings indicate that Sam68 contributes to the establishment of proper innervation and fiber functionality in the developing skeletal muscles, by orchestrating splicing of transcripts involved in neuromaintenance. Neuromuscular defects in *Sam68*$^{-/-}$ mice were correlated with a dramatic loss of motor neurons in the spinal cord

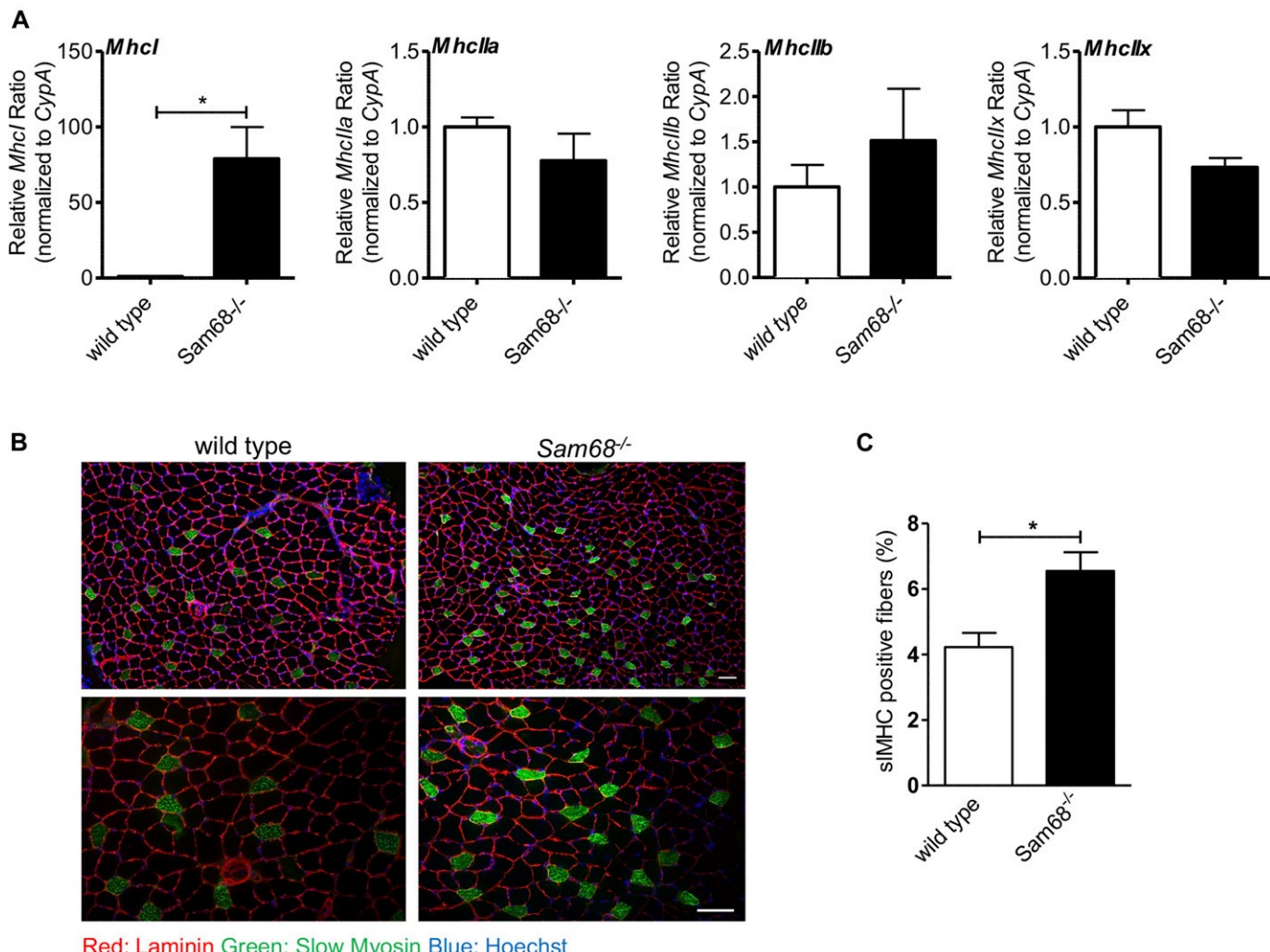

**Figure 6. Sam68<sup>−/−</sup> muscles display alteration in structural composition.**
**(A)** RT-qPCR analysis showing the levels of *MchI*, *MhcIIa*, *MhcIIb*, and *MhcIIx* transcripts in tibialis anterior muscle of 30-d-old control or *Sam68*<sup>−/−</sup> mice normalized for the levels of the housekeeping gene *CypA* (n = 3; means ± SEM); *P*-value was determined by *t* test (*P < 0.05). **(B)** Immunostaining for slow myosin heavy chain (slMHC; green), laminin (red), and Hoechst (blue) of cross section from 30-d-old mice; scale bar: 50 μM; magnification: 10× and 20×. **(C)** Quantification of slMHC-positive muscle fibers in tibialis anterior muscle from 30-d-old mice (n = 3; means ± SEM); *P*-value was determined by *t* test (*P < 0.05).

occurring in the first postnatal month. Sam68 has a predominant nuclear localization in the cell and is strongly expressed in motor neurons (Pagliarini et al, 2015). Our work now shows that Sam68 controls alternative splicing of target transcripts encoding for synaptic proteins in the spinal cord and suggest that dysregulation of this splicing program underlies the defect in muscle innervation and motor neuron survival observed in *Sam68*<sup>−/−</sup> mice.

*Sam68*<sup>−/−</sup> mice are smaller and display reduced body weight compared with wild-type littermates (Richard et al, 2005; Lukong & Richard, 2008). In addition, we found that *Sam68*<sup>−/−</sup> mice display a specific reduction in muscle mass because of reduced size of muscles fibers, whereas their total number was not affected. Notably, this difference dramatically increased from 4 to 12 wk, after the occurrence of a substantial loss of the motor neurons in the spinal cord of *Sam68*<sup>−/−</sup> mice. Loss of motor neurons may result from improper formation of NMJs. In line with this notion, we observed that ablation of Sam68 affects the NMJs structure, with knockout muscles

displaying a higher number of smaller plates than wild type. Changes in NMJ abundance and morphology can be associated with muscle defects. Juvenile rodents have normally a greater NMJ density than adult animals, occupying ~50% of the surface area and 70% of the length, width, circumference, and gutter depth compared with adult muscles (Marques et al, 2000; Ma et al, 2002; Shi et al, 2012; Scurry et al, 2016). Although *Sam68*<sup>−/−</sup> mice display a larger NMJ density per area than wild-type littermates, the general morphology of NMJs in Sam68 KO mice displayed a regular pretzel-like structure, excluding significant defects in muscle growth and indicating the establishment of an atrophic phenotype. Indeed, reduced NMJ size can be also observed in case of muscle atrophy (Scurry et al, 2016). Moreover, the induction of muscle atrophy in *Sam68*<sup>−/−</sup> mice was supported by the up-regulation of atrophy-related genes and the reduction in the number of motor neurons, which can promote muscle atrophy and functional muscle defects. In addition, cytofluorimetric analysis revealed higher percentage of FAPs in the *Sam68*<sup>−/−</sup> muscle, as recently reported

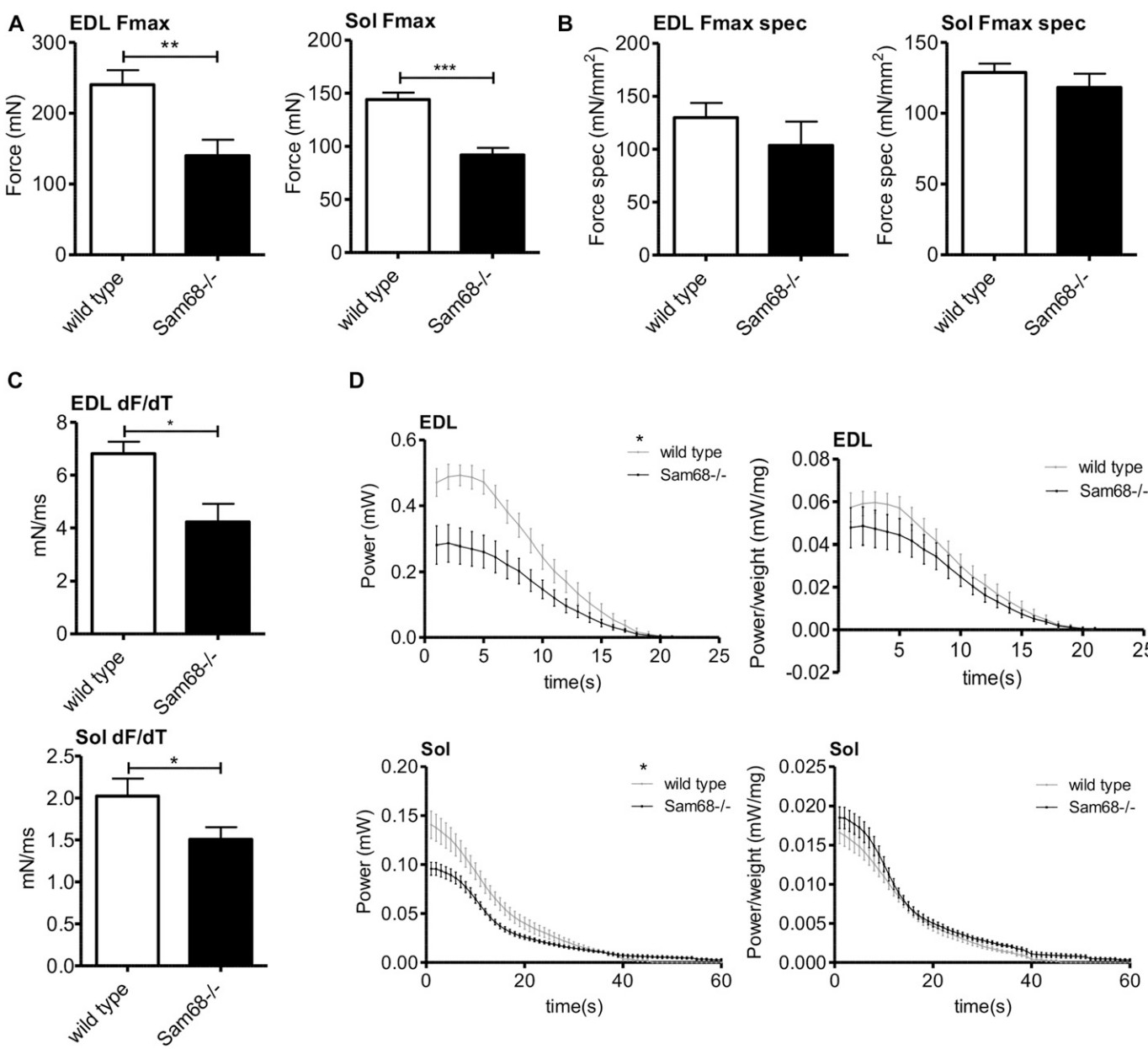

**Figure 7. Sam68 deficiency affects muscle performance.**
**(A, B)** Maximum force and specific maximum force (F/CSA) (B) measured for extensor digitorum longus (EDL) and soleus in isometric conditions. Individual data points represent means ± SEM (n = 8); *P*-value was determined by *t* test (**P < 0.01, ***P < 0.001). **(C)** Speed of force production (dF/dT) measured for EDL and soleus. Individual data points represent means ± SEM; *P*-value was determined by *t* test (*P < 0.05). **(D)** Mechanical power (F*$v_{max}$) and specific power measured for EDL and soleus in isotonic conditions. Individual data points represent means ± SEM (n = 8); two-way ANOVA (*P < 0.05).

in mouse models of neurodegeneration-mediated muscle atrophy (Madaro et al, 2018). These observations support the hypothesis of a denervation-induced condition and suggest that lack of proper neuromuscular connection impairs maturation of the junction and eventually leads to motor neuron degeneration. Thus, impairment of the functional interplay between nerve and muscle can contribute to limit the force-generating capacity of Sam68$^{-/-}$ muscles.

Discrimination between muscle atrophy versus lack of growth of muscle fibers is relatively difficult if only the muscle phenotype is considered. Nevertheless, we addressed this issue by performing various morphological, morphometric, and molecular analyses. Based on our data, we suggest that ablation of Sam68 expression interferes with a series of homeostatic mechanisms, leading to muscle atrophy. In particular, the denervated phenotype, as indicated by a significant proportion of NMJ/bungarotoxin positive element that do not colocalize with the pre-synaptic nerve terminal (synaptophysin), and the reduced number of motor neurons could trigger muscle atrophy, which is promoted by a significant up-regulation of the master-regulatory atrophy genes and, in turn, contributes to the reduced size of Sam68$^{-/-}$ muscle fibers.

Muscle atrophy associated with motor neuron degeneration is often paralleled by alteration in fiber-type composition (Frey et al, 2000; Hegedus et al, 2008; Rocchi et al, 2016). Our data document a strong increase in the expression of type I MHC, a feature of slow-twitch fibers (Castorena et al, 2011), in Sam68$^{-/-}$ TA and EDL muscles and suggest a perturbation in fiber type distribution, which might affect their mechanical properties. In fact, Sam68$^{-/-}$ muscles exhibit a delay in the speed of force production, suggesting that altered Sam68 might confer vulnerability in different muscle groups. Similar mechanical defects were also described in a wobbler mouse model, which also displays a shift from MHC type II toward type I isoforms (Toursel et al, 2000; Agbulut et al, 2004). Furthermore, loss of force generating capacity, muscle atrophy, and MHC transition toward a slow-twitch type has also been reported for the SOD1G93A mouse, a model of amyotrophic lateral sclerosis primary motor neuropathy (Duchen & Strich, 1968; Toursel et al, 2000; Agbulut et al, 2004). These observations suggest that Sam68$^{-/-}$ mice develop a muscular phenotype similar to that observed upon denervation-induced defects in the motor units.

Sam68 is well known for its role in the regulation of alternative splicing (Frisone et al, 2015). Several studies have identified Sam68 target genes that are involved in synaptic transmission, metabolism, and apoptosis (Paronetto et al, 2007, 2011; Iijima et al, 2011; La Rosa et al, 2016; Farini et al, 2020), which may underlie the motor neuron defects observed in our study. Changes in the splicing outcome of the tomosyn-2 gene (Stxbp5l) are particularly relevant for the phenotype. Tomosyn-2 regulates acetylcholine (ACh) secretion at the motor endplate and contributes to overall motor performance (Geerts et al, 2015). Thus, this protein supports the motor unit and enhances synaptic strength during sustained activity to avoid synaptic fatigue upon repetitive stimulation. In this regard, Tomosyn-2$^{-/-}$ mice showed impaired motor performance accompanied by synaptic changes at the NMJ, including enhanced spontaneous acetylcholine release frequency and faster depression of muscle motor endplate potentials during repetitive stimulation (Geerts et al, 2015). Sam68 deficiency affects also the alternative splicing of the AS4 cassette exon in Nrxn1, which is involved in determination of post-synaptic contact (Südhof, 2017). Neurexins are essential for Ca$^{2+}$-triggered neurotransmitter release and Nrxn1-3 knockout mouse models showed impaired neurotransmitter release due to reduction in the synaptic Ca$^{2+}$ channel (Missler et al, 2003). These observations suggest that neurexins organize presynaptic terminals by functionally coupling Ca$^{2+}$ channels to the presynaptic machinery (Missler et al, 2003). These reported findings could partially explain the defects observed in Sam68$^{-/-}$ NMJs. Moreover, we found that Sam68 regulates the alternative splicing of the Gria2 gene in the spinal cord. GRIA2 is a glutamate receptor subunit that controls the Ca$^{2+}$ permeability. Four genes (Gria1-4) encode heteromeric receptors with high affinity for AMPA. Their expression levels, splicing, and mRNA editing lead to differences in Ca$^{2+}$ permeability and gating between cells (Geiger et al, 1995; Jia et al, 1996). Mice defective for either Gria2 mRNA expression or its editing exhibit detrimental phenotypes in synaptic function, development, and behavior (Isaac et al, 2007) and disruption of Gria2 function is associated with neurological disorders such as cerebral ischemia, amyotrophic lateral sclerosis, pain, and epilepsy (Cull-Candy et al, 2006). Notably, Sam68$^{-/-}$ mice display an increase in Gria2 intron 11 retention, which is likely associated with a reduction in the functional receptor (Farini et al, 2020). In addition, we found that Sam68 is required for the

regulation of a set of alternative splicing events encoding post-synaptic proteins in the spinal cord, including collybistin (Arhgef9), gephyrin (Gphn), and densin-180 (Lrrc7). Collectively, the splicing dysregulation of these synaptic genes could explain the impaired integrity of NMJs observed in Sam68$^{-/-}$ mice.

The motor neuron environment determines fiber type composition and muscle performance. Sam68 also modulates the alternative splicing of the ATP-sensitive K(+)-channels (KATP) SUR2, encoded by the Abcc9 gene. This intron retention events lead to a shorter transcript likely targeted to the NMD machinery. A high expression/activity of the sarco-KATP channel is observed in fast-twitch muscles, characterized by elevated muscle strength, whereas a low expression/activity is observed in the slow-twitch muscles characterized by reduced strength and frailty (Tricarico et al, 2016). Thus, the observed switch in fiber composition in Sam68$^{-/-}$ muscles could reflect changes in the expression/activity of sarco-KATP channels. The sarco-KATP channels also play a role in the muscle fatigue. Down-regulation of the KATP subunits of fast-twitch fibers is found in conditions characterized by weakness, frailty and atrophy (Noma, 1983). Muscle fatigue is the decline in force production during prolonged and repetitive stimulation and sarco-KATP channels play a role in reducing resting tension during fatigue (Gong et al, 2000, 2003).

In conclusion, our study highlights an unprecedented link between Sam68 and the integrity of the motor unit and identifies Sam68 as a novel regulator of skeletal muscle properties. In spite of their well-documented impact on several differentiation processes, few RBPs have been studied in skeletal muscle so far. Thus, elucidating the role played by specific RBPs in the development and function of skeletal muscles will likely provide insights into the etiology and pathology of neuromuscular diseases.

# Materials and Methods

### Mice strain

C57/BL6 Sam68 KO mice were generated by replacing exon 4 and part of exon 5 with a neomycin-resistant gene cassette as previously characterized (Richard et al, 2005). Breeding, maintenance, and animal procedures were conducted as described in the project authorized by Ministry of Health (protocol number 510/2017-PR), in accordance with institutional guidelines of the Interdepartmental Service Centre–Station for Animal Technology, University of Tor Vergata, and Fondazione Santa Lucia and in accordance with national and international laws and policies (Directive 2010/63/EU of the European Parliament and of the Council, Italian Legislative Decree 26/2014). Male animals at 4 and 12 wk of age were used for the experiments.

### Flow cytometry analysis of FAPs cells

Hind limb muscles from wild-type mice and Sam68$^{-/-}$ mice were minced and digested in PBS (Sigma-Aldrich) containing 0.1% BSA, 300 µg/ml Collagenase A (Roche), 0.24 U/ml Dispase I (Roche), 2 µg/ml DNase I (Roche), 50 µM CaCl$_2$, and 1 mM MgCl$_2$ for 60 min at 37°C under constant agitation. Digested muscle cells were stained with

primary antibodies (1:50) CD31-eFluor450 (eBioscience), CD45-eFluor450 (eBioscience), Ter119-eFluor450 (eBioscience), Sca-1-FITC (BD Pharmingen), and (1:500) α7integrin-APC (AbLab) for 30 min at RT. Cells were finally washed and resuspended in PBS, 0.1% sodium azide, and 0.2% FBS. Flow cytometry was performed with a MoFlo High Speed Cell Sorter (Beckman Coulter) and analysis using FlowJo-10 software. FAP cells were identified as Ter119–/CD45–/CD31–/α7-integrin⁻/Sca-1⁺ cells.

## Isolation of total RNA and conventional and quantitative PCR

Total RNA was extracted by using TRIzol reagent (Life Technologies) according to the manufacturer's instructions. RNA was subjected to DNAse digestion (QIAGEN), and the first-strand cDNA was obtained from 1 $\mu g$ of RNA using random primers and M-MLV reverse transcriptase (Promega). Synthesized cDNA from total RNA was used for conventional PCR (GoTaq G2; Promega) and quantitative PCR (SYBR Green Master Mix for Light-Cycler 480; Roche), according to the manufacturer's instructions. Primers used for qPCR and PCR analyses are listed in Table S1.

For miRNA expression analysis, the TaqMan method was used. 20 ng of total RNA was reverse-transcribed using TaqMan miRNA Reverse Transcription Kit (4366596; Applied Biosystems) following the manufacturer's instructions. Then 1.5 $\mu l$ of each miR-specific cDNA was submitted to PCR amplification by using TaqMan universal PCR master mix II (4440044; Applied Biosystems). The following TaqMan miRNA assays were used as probes: hsa-miR-23a (000399) and U6 snRNA (001973). *Cyclophilin A*, *Gapdh*, 18S, or U6 snRNA were used as internal controls.

## Protein extraction and Western blot analyses

Protein extracts were prepared using radio-immuno-precipitation assay buffer supplemented with 1 mM dithiothreitol, 10 mM $\beta$-glycerophosphate, 1 mM $Na_3VO_4$, 10 mM NaF, and protease inhibitor cocktail (Sigma-Aldrich). The protein extracts were incubated on ice for 10 min and then centrifuged for 10 min at 12,000$g$ at 4°C. Protein quantification was performed by Quick Start Bradford Protein Assay (Bio-Rad). Cell extracts were diluted in Laemmli buffer and boiled for 5 min at 95°C. Extracted proteins (30–50 $\mu g$) were separated on 10% SDS–PAGE gels and transferred to Hybond-P membranes (GE Healthcare). Membranes were saturated with 5% non-fat dry milk in PBS containing 0.1% Tween-20 for 1 h at RT and incubated with the following antibodies overnight at 4°C: rabbit anti-Sam68 1:1,000, rabbit anti-MuRF1 1:200, rabbit anti–atrogin-1 1:200, mouse anti-GAPDH 1:1,000, and mouse anti-$\beta$ ACTIN 1:1,000 (all from Santa Cruz). Secondary antimouse or antirabbit IgGs conjugated to horseradish peroxidase (Amersham) were incubated with the membranes for 1 h at RT at a 1:10,000 dilution in PBS containing 0.1% Tween-20. Immunostained bands were detected by a chemiluminescent method (Thermo Fisher Scientific). Densitometric analysis was obtained by ImageJ software.

## Histology and immunohistochemistry

TA muscles from 4- to 12-wk-old wild-type and Sam68⁻/⁻ mice were conserved in tissue-freezing medium and snap-frozen in liquid nitrogen–cooled isopentane. For morphometric analysis, transversal cryostat sections were stained with hematoxylin and eosin (H&E) according to the standard protocols. Images were obtained using Axioplan microscope (Carl Zeiss Microimaging, Inc.) and processed using Axiovision software (V 4.8.2.0). A minimum of two muscle sections, arbitrarily chosen from the middle region of each muscle (n = 3 per group), were analyzed with ImageJ software (v.1.51j8; National Institutes of Health) to quantify the total number of fibers per muscle section. The CSA of single myofibers was quantified by analyzing a minimum of six muscle sections, arbitrarily chosen from the entire muscle (n = 3 per group). Immunofluorescence analysis were performed on TA muscles from wild-type and Sam68⁻/⁻ mice at 4 wk of age. 12-$\mu$m-thick cryostat sections were immunostained using Anti-Laminin (L9393; Sigma-Aldrich), Monoclonal Anti-Myosin Slow (M8421; Sigma-Aldrich), and appropriate fluorescent secondary antibodies (A-11011; A-11001; Invitrogen). Hoechst staining was used to visualize nuclei. Three transversal sections from the middle region of each muscle were photomicrographed (n = 3 mice/genotype); images were obtained using Axio Imager A2 microscope (Carl Zeiss Microimaging, Inc.) and processed by ZEN2 software (Blue edition). The percentage of slow myosin–positive (SlMHC+) fibers was calculated by [SlMHC+]/[Total Fibers] per tissue section.

For Sam68 immunofluorescence analyses, frozen sections (6 $\mu m$ thick) were immunostained using Anti-Sam68 (SC-333; Santa Cruz) and Anti-Laminin (ALX-804-190-C100) primary antibodies and appropriate fluorescent secondary antibodies (A-11034; A-11007; Invitrogen). Nuclei were detected by Hoechst staining.

For NMJ analysis, the TA muscle was dissected and fixed in 4% PFA at 4°C for 180 min. Small bundles of muscle fibers were isolated under a dissecting microscope and immunostained with mouse antineurofilament (1:200; SMI-312; BioLegend) and rabbit anti-synaptophysin (1:200; Thermo Fisher Scientific). Neurofilaments were visualized with TRITC AP donkey antimouse IgG (1:200; Jackson ImmunoResearch Laboratories, Inc.), and synaptic vesicles were visualized with Cy5 AP donkey antirabbit IgG (1:400; Jackson ImmunoResearch Laboratories, Inc.) secondary antibody. AChRs were labeled with Alexa Fluor 488–conjugated $\alpha$-bungarotoxin (10 nM; Molecular Probes). Z-stack images were obtained at sequential focal planes 3 $\mu m$ apart using a confocal microscope (Laser Scanning TCS SP2; Leica). NMJs were analyzed in terms of number per field and area of the individual endplate using LAS AF Lite software (Leica). For each genotype, a minimum of 65 optical sections and 300 endplates were evaluated from randomly selected visual fields. Blind acquisition and analysis were performed using coded slides from three animals for each genotype. Representative images are flattened projections of Z-stack images.

For ChAT and Neurotrace double-staining, mice were anaesthetized with Rompun (20 mg/ml, 0.5 ml/kg, i.p.; Bayer) and Zoletil (100 mg/ml, 0.5 ml/kg; Virbac) and perfused transcardially with 50 ml saline followed by 50 ml of 4% paraformaldehyde in PB (0.1 M, pH 7.4). Spinal cords were removed and post-fixed in paraformaldehyde at 4°C and then immersed in 30% sucrose solution at 4°C until sinking. Coronal sections of the lumbar spinal cord (L1–L5) were cut with a cryostat at 30-$\mu$m thickness. The selected sections were processed with the primary anti-ChAT antibody in PB containing Triton 0.3% overnight. After three washes in PB, the sections were

immunostained using Anti-ChAT (AB144P; Millipore), NeuroTrace (N21482; Thermo Fisher Scientific), and appropriate fluorescent secondary antibodies donkey antigoat IgG (A32814; Thermo Fisher Scientific).

## Motor neuron count in lumbar spinal cord

Spinal cords were obtained as previously described (see Histology and immunohistochemistry section). Coronal sections of the lumbar spinal cord (L1–L5) were cut with a cryostat at 30-$\mu$m thickness, and every seventh section was stained with Nissl substance (n = 3 animal/group; n = 6 slices/animal). One Nissl-stained section every two and one every six were used at P8 and P30, respectively. Motoneurons were clearly recognized for their large size, for their intensely Nissl-stained cytoplasm, and for their prominent nucleolus (Guo et al, 2013). An optical fractionator stereological design (West et al, 1991) was used to obtain impartial estimates of the total number of motor neurons using the Stereo Investigator system (Stereo Investigator software, version 4.04; MicroBrightField). A stack of MAC 5000 controller modules (Ludl Electronic Products) was configured to interface with a microscope (BX 50; Olympus) with a motorized stage and a color digital camera (HV-C20; Hitachi) with a Pentium II PC workstation. A 3D optical dissector counting probe (x, y, and z dimensions of 30 × 30 × 10 $\mu$m, respectively) was applied to a systematic random sample of motor neurons in the lumbar spinal cord. The region of interest was outlined using the 10× objective, whereas the 100× oil immersion objective was used for marking individual motor neurons. The total cell number was estimated according to the formula:

$$N = SQ \times \frac{1}{ssf} \times \frac{1}{asf} \times \frac{1}{tsf}$$

where SQ is the number of neurons counted in all optically sampled fields of the area of interest, ssf is the section sampling fraction, asf is the area sampling fraction, and tsf is the thickness sampling fraction.

## Histological analysis of sciatic nerve

For the histological analysis, 8-$\mu$m nerve cryosections were analyzed. Cryosections and cultured cells were fixed in 100% acetone for 1 min at RT. Nerve sections were then blocked for 1 h with a solution containing 4% BSA in PBS. Neurofilament (NF-L) staining was performed by an antigen retrieval protocol. Primary antibodies (neurofilament and laminin) immunostaining was performed O/N at 4°C and then the antibody binding specificity was revealed using secondary antibodies coupled to Alexa Fluor 488 or 594 (Invitrogen). Sections were incubated with DAPI in PBS for 5 min for nuclear staining, washed in PBS, and mounted with mounting medium or glycerol (3:1 in PBS). The primary antibodies used for immunofluorescences are rabbit anti-laminin (#L9393, 1:400; Sigma-Aldrich) and mouse anti-neurofilament (#sc-20012, 1:100; Santa Cruz). The figures reported are representative of all the examined fields.

## Laser-capture microdissection

Spinal cords were obtained from P45 male mice, included in OCT compound (VWR), frozen in powdered dry ice, and stored at −80°C. 10 $\mu$m frozen sections cut on a cryostat (Leica CM1850) were mounted on PET membrane of 1.4-$\mu$m frame slides (Leica) previously cleaned with RNase (Molecular Bio Products) and UV-treated for 45 min under sterile hood. Modified cresyl violet staining for RNA research (0.5 g cresyl violet into 50 ml 100% ethanol) was performed to visualize the neural structure. The selected area was microdissected with a laser-microdissection system (Leica LMD6) and recovered in RNAlater reagent (QIAGEN). Total RNA was extracted from the dissected specimen using an RNAeasy Micro Kit (QIAGEN) and quantified with Agilent Bioanalyzer 2100 using RNA600 picoKit. cDNA was reverse-transcribed using SuperScript-IV VILO master mix with EZ DNase (Invitrogen).

### Strength test

EDL and soleus muscles were excised from the animal and kept immersed in a Krebs–Ringer bicarbonate buffer (K4002; Sigma-Aldrich) solution added with potassium phosphate (1.2 mM), magnesium sulfate (0.57 mM), calcium chloride (2.00 mM), and Hepes (10.0 mM) and gassed with a mixture of 95% $O_2$ and 5% $CO_2$ at RT. Muscles were mounted vertically in a temperature-controlled (30°C) chamber. One end of the muscle was linked to a fixed clamp, whereas the other end was connected to the lever arm of an Aurora Scientific Instruments 300B actuator/transducer system, using a nylon thread. The isolated muscle was electrically stimulated by means of two platinum electrodes, located 2 mm from each side of the muscle, with 200 mA controlled current pulses. Both muscles were stimulated with a single pulse to measure the contraction kinetics, whereas for the other tests, the muscle was stimulated with pulse trains at tetanic frequency; 180 Hz for EDL and 80 Hz for soleus, respectively (Del Prete et al, 2008).

## Hanging-wire test

Neuromuscular strength was tested by the hanging-wire test. Each mouse was placed on a wire lid of a conventional housing cage and the lid was turned upside down. The latency from the beginning of the test until the mouse stood with at least two limbs on the lid was timed. The animals had three attempts to stand for a maximum of 180 s per trial, and the longest latency was recorded (Oliván et al, 2015).

## Statistical analyses

All data are expressed as the mean ± SEM as indicated in the figure legends. Two-tailed $t$ test and one-way or two-way ANOVA were performed using Prism 5 software (GraphPad Software).

# Supplementary Information

# Acknowledgements

This work was supported by grants from the Associazione Italiana Ricerca sul Cancro (AIRC) (IG21877) to MP Paronetto and from Ministry of Health "Ricerca

Finalizzata" (RF-2016-02363460 and RF-2013-02358910) to C Sette and A Musarò. Ministry of University and Research (MIUR, PRIN 2017) to C Sette. N Mercatelli was supported by a scholarship from Fondazione Umberto Veronesi. L Madaro was supported by the Italian Ministry of Health (grant no. GR-2013-02356592). The authors wish to thank Davide BONVISSUTO for technical support in the laser microdissection procedure.

## Author Contributions

E De Paola: data curation, formal analysis, validation, investigation, and methodology.
L Forcina: data curation, formal analysis, investigation, and methodology.
L Pelosi: investigation and methodology.
S Pisu: data curation, investigation, and methodology.
P La Rosa: investigation and methodology.
E Cesari: methodology.
C Nicoletti: methodology.
L Madaro: investigation, visualization, methodology, and writing—review and editing.
N Mercatelli: methodology.
F Biamonte: methodology.
A Nobili: methodology.
M D'Amelio: data curation, investigation, methodology, and writing—review and editing.
M De Bardi: methodology.
E Volpe: methodology.
D Caporossi: conceptualization and writing—review and editing.
C Sette: conceptualization, supervision, funding acquisition, and writing—original draft, review, and editing.
A Musarò: conceptualization, data curation, supervision, funding acquisition, and writing—original draft, review, and editing.
MP Paronetto: conceptualization, data curation, supervision, funding acquisition, investigation, project administration, and writing—original draft, review, and editing.

## Conflict of Interest Statement

The authors declare that they have no conflict of interest.

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
