## [Reviewer comments · Life Science Alliance]

Life Science Alliance

Sam68 splicing regulation contributes to motor unit establishment in the postnatal skeletal muscle

Elisa De Paola, Laura Forcina, Laura Pelosi, Simona Pisu, Piergiorgio La Rosa, Eleonora Cesari, Carmine Nicoletti, Luca Madaro, Neri Mercatelli, Filippo Biamonte, Annalisa Nobili, Marcello D'Amelio, Marco De Bardi, Elisabetta Volpe, Daniela Caporossi, Claudio Sette, Antonio Musarò, and Maria Paronetto

DOI: <https://doi.org/10.26508/lsa.201900637>

Corresponding author(s): *Maria Paronetto, University of Rome Foro Italico*

Review Timeline:

Submission Date:	2019-12-27
Editorial Decision:	2020-01-16
Revision Received:	2020-07-08
Editorial Decision:	2020-07-17
Revision Received:	2020-07-23
Accepted:	2020-07-24

Transaction Report:

January 16, 2020

Re: Life Science Alliance manuscript #LSA-2019-00637-T

Prof. Maria Paola Paronetto
University of Rome Foro Italico
Department of Movement, Human and Health Sciences
Piazza Lauro de Bosis, 15
Rome, Italy 00135
Italy

Dear Dr. Paronetto,

Thank you for submitting your manuscript entitled "Sam68 splicing regulation insures proper motor unit establishment in the postnatal skeletal muscle" to Life Science Alliance. The manuscript was assessed by expert reviewers, whose comments are appended to this letter.

As you will see, your work received a somewhat split view. While reviewer #1 raises only concerns that seem straightforward to address, reviewer #2 and #3 expect more insight into the potential causality of motoneuron loss/Sam68-dependent changes in transcripts for the observed muscle phenotype. The reviewers also raise other concerns and point out that some of your conclusions are not sufficiently supported at this stage.

Based on the input received, we concluded that we can offer to consider a revised version for publication here, should you be able to address the reviewer concerns. We would thus like to invite you to submit a revised version of your manuscript to us. We think it may be most productive to discuss the revision further. Therefore, once you've had the time to evaluate all reviewer comments, we would welcome a preliminary point-by-point response in which you outline how you could/would address the issues raised.

When submitting the revision, please include a letter addressing the reviewers' comments point by

point.

Thank you for this interesting contribution to Life Science Alliance. We are looking forward to receiving your revised manuscript.

Sincerely,

B. MANUSCRIPT ORGANIZATION AND FORMATTING:

Reviewer #1 (Comments to the Authors (Required)):

Sam68 splicing regulation insures proper motor unit establishment in the postnatal skeletal muscle (Elisa De Paola et al.)

The authors of this study identify the RNA binding protein Sam68 as an important player in maintaining the integrity of the NMJ, with subsequent effects on skeletal muscle function. Alternative splicing events of key targets are observed in the spinal cords of Sam68-ko mice. Previous studies have shown that these targets are involved in synaptic transmission, metabolism, and apoptosis, and thus have relevance for the observed phenotypes of muscle atrophy, muscle performance, and loss of motor neurons. Furthermore, it is shown that Sam68-ko mice display a switch in muscle fiber type from fast-twitch to predominantly slow-twitch, a phenomenon that is often observed simultaneously with motor neuron degeneration and muscle atrophy. Overall, the authors present a clear study which highlights a new role for Sam68 in skeletal muscle function.

Minor revisions

Figure 1:

1. Rationale for selected targets should be more clearly explained. It is clear in Fig 1 that they are all alternatively spliced in the Sam68-/- mice and are all related to the phenotype, but from where were they chosen? RNAseq data?
2. The functional significance of the transcripts selected for alternative splicing analysis is explained in the discussion, but it might be easier for the reader if the authors would explain in this section that, for example, Sgce has been indicated in muscular dystrophies and thus has a connection to the observed phenotype in Sam68-/- mice.
3. While the Sam68-ko mouse strain is well established, a validation of the knockout should be shown here (or in supplemental) with qPCR or WB from the spinal cord.

Figure 2:

4. In panel A, clarify how the top and bottom row are different. Is it magnified?
5. The authors should indicate whether having more NMJs with smaller areas is a known indicator of weaker muscle strength.
6. Number of NMJs was quantified at roughly 4 weeks of age (panel B), which is when total number of motor neurons was observed to drastically decreased (panel F). How are there more NMJs at 4 weeks, when motor neuron numbers are in decline?
7. Add error bars for wildtype in panel G.

Figure 5:

8. qPCR should be normalized to more than one housekeeping gene (MIQE).
8. Panel B and C are switched in the figure legend

Figure 6:

9. Hindlimb muscles have different proportions of slow and fast twitch fibers, the TA is mostly fast twitch and was the muscle selected for these experiments. It would also be informative to show analysis for a predominantly slow twitch muscle, such as the soleus, to observe whether there are still more slow twitch fibers with unchanged levels of fast twitch fibers in the Sam68-ko.

Discussion:

10. The authors state "Our data document a strong increase in the expression of type I MHC both at the RNA and at the protein level in Sam68-/- TA and EDL muscles..." but data is not shown for EDL muscles.

Reviewer #2 (Comments to the Authors (Required)):

In the submitted work, De Paola and colleagues analysed the motoneuronal and muscular phenotype of a constitutive KO of the RNA binding protein (RBP) Sam68. The major conclusion of this study is that aberrant splicing regulation of genes involved in synaptic function in the spinal cord leads to loss of motoneurons and consequent muscular atrophy.

General comment:

The histological characterization of the muscular phenotype of the KO mice is in principle interesting and well performed, even if open to amelioration (increasing the number of repetitions and type of experiment, see below). On the other side, the novelty of the work is rather limited from a molecular point of view. It is already known that Sam68 regulates the splicing of the genes analysed and in this work there is no experiment that explores the mechanistic role of some of the investigated transcripts in the observed phenotype. Moreover, the conclusion that the muscular phenotype is entirely dependent on the motoneuronal loss is questionable. Is Sam68 expressed in the muscles? What about regulation of splicing of specific muscular transcripts? This has not been investigated. In conclusion, this work presents some novelty regarding the role of Sam68 in the neuromuscular junction (NMJ) formation or maintenance but does not advance the knowledge on the molecular function of Sam68. A further effort in this direction would improve the work.

Other specific comments:

- 1) Several experiments have been performed on only 3 independent samples. This number is rather low and conclusions about statistical significance can lead to wrong conclusions. It is advisable to increase the n.
- 2) In several cases, splicing alterations or expression levels have been analysed using RT-PCR and not real-time PCR. This does not allow conclusive quantification.
- 3) In Figure 1 the authors analyse alternative splicing for a number of target genes. How did they select these transcripts? Are all these transcripts already validated targets of Sam68? Is this just a confirmatory study? What is the effect at the protein levels of these splicing alterations? A western blot showing the effect at the protein level would be required.
- 4) Experiments have been performed on RNA extracted from spinal cords which contain motoneurons but also several other type of neurons. The high difference between WT and KO mice could be due to a regulation common to other neurons of the spinal cord. To claim that these transcripts are coming from motoneurons, the authors should confirm their results on isolated motoneurons.
- 5) The study explores in detail the muscular phenotype, which is suggested to be dependent on motoneuronal loss. However, this has been explored only superficially. Authors claim KO mice at 1 week present intact motoneurons but data are not shown. Quantification of motoneurons has been performed on Nissl-stained slices. Other markers as ChAT or Hb9 would be more appropriate. How many sections per mouse have been quantified? Details should be specified. In addition, the authors should analyse the sciatic nerves to confirm signs of degeneration.
- 6) Figure 2A. The authors combine three antibodies, two of which with a fluorophore in the same channel that cannot be distinguished (SMI31 and synaptophysin). This is not good practice and does not really allow to interpret the results. In the Results, the text describing this experiment is confusing.

Reviewer #3 (Comments to the Authors (Required)):

In this manuscript, De Paola et al study the impact of the RNA binding protein Sam68 on skeletal muscle. They used a Sam68 knock out mouse model and revealed that Sam68 is essential for proper neuromuscular connections and muscle strength. The data highlight a high alteration of the neuromuscular system in Sam68^{-/-} that will be important for the field. However, in my opinion analysis-interpretations are sometimes imprecise and their completion would strengthen the manuscript.

Major points

1. The title of the manuscript is misleading "Sam68 splicing regulation insures proper motor unit establishment ...". Unless the authors make a causal link between alternative splicing dysregulations in Figure 1 and the alteration of the neuromuscular system in the rest of the paper, the title should be modified.
2. Figure 2A, B: In Sam68^{-/-} mice (and to a lesser extent in WT), there are a major proportion of "NMJ"/ bungarotoxin positive element that do not colocalize with the pre-synaptic nerve terminal (synaptophysin). Could it be an indication for initiation of NMJ disruptions at 30dpp? If this is the case, the quantification in Figure 2B should distinguish the different types / degeneration stages of NMJs.
3. In general, it would be important to look at the muscle fiber, the motor neuron and their junctions at earlier time points. It would give some indication in regards to the developmental stages from when Sam68^{-/-} phenotypes are expressed. It could also help in identifying the first - and perhaps the causal -element leading to the substantial muscle defects. It is of particular relevance since in their previous study (Pagliarini et al, 2015), the authors showed that ablation of Sam68 do not affect NMJs and muscle fiber sizes at 10 dpp (in non SMA mice).
4. I found sometimes difficult to make the distinction between atrophy versus lack of growth of muscle fibers (Figures 3 and 4). The authors should be more cautions with respect to their conclusions or better support them experimentally.
5. The authors mention that Sam68 is highly expressed in the motor neurons of spinal cord and revealed some substantial dysregulations of alternative splicing in Sam68^{-/-} spinal cord. Is the ablation of Sam68 in motor neurons responsible of the skeletal muscle alteration?

Minor points

1. Figure1A: The authors claim that Sam68 in spinal cord favors skipping of Nrnx3 exon20. However, the PCR signal appears to be saturated preventing a rigorous quantification. The authors should use an accurate quantitative method to analyze Nrnx3 exon20 alternative splicing in Sam68^{-/-}.
2. Figure 2E: i) please show the whole spinal cord section for both WT and KO mice and ii) label the images.
3. Figure 3C, D: please add the age of the animals on the figure as well.
4. Figure 5: The authors should ensure FoxO3A and Myogenin are increased at the protein level in Sam68^{-/-} to support their conclusion.

5. Figure 7: I could not find at which age the experiments were performed.

Reviewer #1 (Comments to the Authors (Required)):

Sam68 splicing regulation insures proper motor unit establishment in the postnatal skeletal muscle (Elisa De Paola et al.)

The authors of this study identify the RNA binding protein Sam68 as an important player in maintaining the integrity of the NMJ, with subsequent effects on skeletal muscle function. Alternative splicing events of key targets are observed in the spinal cords of Sam68-ko mice. Previous studies have shown that these targets are involved in synaptic transmission, metabolism, and apoptosis, and thus have relevance for the observed phenotypes of muscle atrophy, muscle performance, and loss of motor neurons. Furthermore, it is shown that Sam68-ko mice display a switch in muscle fiber type from fast-twitch to predominantly slow-twitch, a phenomenon that is often observed simultaneously with motor neuron degeneration and muscle atrophy. Overall, the authors present a clear study which highlights a new role for Sam68 in skeletal muscle function.

Minor revisions

Figure 1:

1. Rationale for selected targets should be more clearly explained. It is clear in Fig 1 that they are all alternatively spliced in the Sam68^{-/-} mice and are all related to the phenotype, but from where were they chosen? RNAseq data?

*The alternative splicing changes selected derive from the analysis of several RNA sequencing and microarray datasets of experiments using Sam68-depleted cells or Sam68^{-/-} mouse tissues, which were performed by our and other groups (Chawla et al., 2009; Paronetto et al., 2011; Witte et al., 2019; Farini et al., 2020). This information is now provided in the revised manuscript at **page 6**.*

2. The functional significance of the transcripts selected for alternative splicing analysis is explained in the discussion, but it might be easier for the reader if the authors would explain in this section that, for example, Sgce has been indicated in muscular dystrophies and thus has a connection to the observed phenotype in Sam68^{-/-} mice.

As suggested, the functional significance of the selected alternative splicing changes has now been explained in the results section when they are first described.

3. While the Sam68-ko mouse strain is well established, a validation of the knockout should be shown here (or in supplemental) with qPCR or WB from the spinal cord.

As requested, we now provide evidence of Sam68 ablation in the spinal cord by qPCR analysis (Supplementary Figure 1A). Due to paucity of material in the spinal cord, we show Western blot

analysis of skeletal muscle extracts from the same mice for validation of the knockout at the protein level.

Figure 2.

4. Figure 2: In panel A, clarify how the top and bottom row are different. Is it magnified?

Different magnification has been used for the two pictures. This issue is now more clearly indicated in the Figure legend. To help the proper interpretation of the representative images, we added also a 50 μm scale bar on both panels.

5. The authors should indicate whether having more NMJs with smaller areas is a known indicator of weaker muscle strength.

We thank the Reviewer for this suggestion. We improved the Discussion section about the analysis of NMJs by indicating that the alteration of NMJ abundance and morphology could be associated with muscle defects. In particular, we state that juvenile rodents have normally a greater NMJ density compared with adult animals, occupying approximately 50% of the surface area and 70% of the length, width, circumference, and gutter depth compared with adult muscles (Ma J et al, 2002; Scurry et al 2016; Shi et al.2012; Marques 2000). Although the Sam68ko mice display a larger NMJ density per area compared to wild type littermates, the general morphology of NMJs displayed a regular pretzel-like structure, excluding significant defects in muscle growth and indicating the establishment of an atrophic phenotype. Indeed, NMJs with a reduced size can be also observed in case of muscle atrophy (Chuna et al 2019; Scurry et al 2016). The promotion of muscle atrophy in Sam68ko mice is also supported by molecular and morphometric analysis, which reveal an up-regulation of atrophy-related genes and the reduction in the number of motor neurons, which can promote muscle atrophy and functional muscle defects. Thus, motor neuron impairment, reduced muscle mass, and alteration in NMJ homeostasis can significantly affect the force-generating capacity of skeletal muscle.

6. Number of NMJs was quantified at roughly 4 weeks of age (panel B), which is when total number of motor neurons was observed to drastically decreased (panel F). How are there more NMJs at 4 weeks, when motor neuron numbers are in decline?

The higher density of NMJ in Sam68^{-/-} mice can be associated with the alteration in the maturation processes of NMJ, potentially due to defect in muscle innervation. We observed a significant proportion of NMJ/ bungarotoxin positive element that do not colocalize with the pre-synaptic nerve terminal (synaptophysin). This can be an indication of reduced muscle innervation, due to reduced number of motor neurons.

7. Add error bars for wildtype in panel G.

Error bars have been added as suggested.

8. Figure 5: qPCR should be normalized to more than one housekeeping gene (MIQE).

The qPCR analyses were normalized for Gapdh, Cyclophilin A and 18S, as indicated in the methods.

Figure 6: Panel B and C are switched in the figure legend Figure 6.

This mistake has now been corrected. We thank the Reviewer for pointing it out.

9. Hindlimb muscles have different proportions of slow and fast twitch fibers, the TA is mostly fast twitch and was the muscle selected for these experiments. It would also be informative to show analysis for a predominantly slow twitch muscle, such as the soleus, to observe whether there are still more slow twitch fibers with unchanged levels of fast twitch fibers in the *Sam68*^{-/-}.

We have performed the analysis of CSA and myosin isoforms expression in TA, EDL and soleus as prototype of mixed, slow and fast twitch muscle, respectively; this information is included in the revised manuscript (Supplementary Figure 7).

Discussion:

10, The authors state "Our data document a strong increase in the expression of type I MHC both at the RNA and at the protein level in *Sam68*^{-/-} TA and EDL muscles..." but data is not shown for EDL muscles.

As mentioned above, qPCR analysis of MHC isoforms in the EDL and Soleus muscles are now included in the Supplementary Figure 7 of the Revised manuscript.

Reviewer #2 (Comments to the Authors (Required)):

In the submitted work, De Paola and colleagues analysed the motoneuronal and muscular phenotype of a constitutive KO of the RNA binding protein (RBP) Sam68. The major conclusion of this study is that aberrant splicing regulation of genes involved in synaptic function in the spinal cord leads to loss of motoneurons and consequent muscular atrophy.

General comment:

The histological characterization of the muscular phenotype of the KO mice is in principle interesting and well performed, even if open to amelioration (increasing the number of repetitions and type of experiment, see below). On the other side, the novelty of the work is rather limited from a molecular point of view. It is already known that Sam68 regulates the splicing of the genes analysed and in this work there is no experiment that explores the mechanistic role of some of the investigated transcripts in the observed phenotype.

Moreover, the conclusion that the muscular phenotype is entirely dependent on the motoneuronal loss is questionable.

We agree with the Reviewer that our study does not prove that the muscular phenotype is "entirely dependent on motor neuron loss". Indeed, in the original version of the manuscript we tried to stress the concept that motor neuron loss can explain many aspects of the phenotype we observe. This conclusion has now been rephrased more clearly in the revised manuscript to avoid confusing claims (see page 15).

Is Sam68 expressed in the muscles? What about regulation of splicing of specific muscular transcripts? This has not been investigated.

Yes, Sam68 is expressed in muscle (see the western blot analysis in Supplementary Figure 1B). However, no muscle-specific targets of Sam68 are known. As splicing is tissue-specific, we would need to perform RNA sequencing to identify them. However, this type of experiment would require time beyond the frame of this revision, which was already delayed due to the COVID-related lockdown we experienced in our country. In the revised manuscript, we partially addressed this issue by toning down our conclusions (see page 15).

In conclusion, this work presents some novelty regarding the role of Sam68 in the neuromuscular junction (NMJ) formation or maintenance but does not advance the knowledge on the molecular function of Sam68. A further effort in this direction would improve the work.

Other specific comments:

1) Several experiments have been performed on only 3 independent samples. This number is rather low and conclusions about statistical significance can lead to wrong conclusions. It is advisable to increase the n.

As requested, in the revised manuscript we now provide data from 5 independent samples.

2) In several cases, splicing alterations or expression levels have been analysed using RT-PCR and not real-time PCR. This does not allow conclusive quantification.

Some of the Sam68-regulated splicing events have a complex pattern and other were in very small exon, thus making design of qPCR primers very troublesome. When possible by primers design, we validated the splicing changes by real-time PCR as requested. The new data are shown in Fig 1, and Supplementary Figure 2.

3) In Figure 1 the authors analyse alternative splicing for a number of target genes. How did they select these transcripts? Are all these transcripts already validated targets of Sam68? Is this just a confirmatory study? What is the effect at the protein levels of these splicing alterations? A western blot showing the effect at the protein level would be required.

The alternative splicing changes selected derive from the analysis of several RNA sequencing and microarray datasets of experiments using Sam68-depleted cells or Sam68^{-/-} mouse tissues, which were performed by our and other groups (Chawla et al., 2009; Paronetto et al., 2011; Witte et al., 2019; Farini et al., 2020). Some of them were already validated targets, whereas other targets were validated in this work for the first time. This information is now provided in the revised manuscript at page 6. Importantly, since splicing regulation is highly cell- and tissue-specific, validation of these targets in the spinal cord provides insightful information.

Unfortunately, antibodies specific for the splice variants investigated in our study were not readily available.

4) Experiments have been performed on RNA extracted from spinal cords which contain motoneurons but also several other type of neurons. The high difference between WT and KO mice could be due to a regulation common to other neurons of the spinal cord. To claim that these transcripts are coming from motoneurons, the authors should confirm their results on isolated motoneurons.

To answer this criticism, we have performed laser microdissection of motoneurons from the anterior horns of the spinal cord (see new Supplementary Figure 3A). Taking advantage of this technique, we were able to validate alternative splicing changes also in isolated motor neurons (Supplementary Figure B-F).

5) The study explores in detail the muscular phenotype, which is suggested to be dependent on motoneuronal loss. However, this has been explored only superficially. Authors claim KO mice at 1week present intact motoneurons but data are not shown. Quantification of motoneurons has been performed on Nissl-stained slices. Other markers as ChAT or Hb9 would be more

appropriate. How many sections per mouse have been quantified? Details should be specified. In addition, the authors should analyse the sciatic nerves to confirm signs of degeneration.

As requested, we now show also data from mice at 8 days post partum (dpp) and report no significant changes in motor neuron counts at this stage of development (new Fig. 2G). Nissl-staining has been used in several studies for this purpose, because motor neurons can be easily recognized given the size of their cell body (Vercelli et al, 2008; Guo et al, 2013). As suggested, we now show in the Supplementary Figure 5 that Nissl-stained (Neurotrace, which is red fluorescent Nissl) neurons are also ChAT-positive.

The requested methodological information has been provided in the revised manuscript.

Moreover, to confirm motor neuron degeneration we analyzed the sciatic nerves of wild type and knockout mice, as suggested by the Reviewer. These new experiments have been included in the Revised Figure 2H-I and show a significant increase in motor neuron axonal degeneration in the sciatic nerve.

6) Figure 2A. The authors combine three antibodies, two of which with a fluorophore in the same channel that cannot be distinguished (SMI31 and synaptophysin). This is not good practice and does not really allow to interpret the results. In the Results, the text describing this experiment is confusing.

In the revised manuscript (legend of Supplementary Figure 4) we better clarify that the immunofluorescence analysis on NMJs was performed using three fluorophores, each one lying in a different channel during the confocal microscopy acquisition. In particular, a 488-conjugated bungarotoxin (emission maximum 520 nm) was used to label the nAChR clusters at the NMJ; the pre-synaptic terminal was visualized through a Cyanine 5 secondary antibody (Cy5_emission maximum 669 nm) to detect synaptophysin and a Tetramethylrhodamine-isothiocyanate secondary antibody (TRITC_emission maximum 569 nm) to detect neurofilament markers. We decided, as a representative and well-established convention, to assign the red colour to both Cy5 and TRITC channel in order to easily discern the pre-synaptic terminal from the post-synaptic plaque. Moreover, we edited the description of the experiment in the Results section and we better clarified the technical details in the section of Methods. Furthermore, we now show in the Supplementary Figure 4 the single channel-pictures for the immunoistochemical analyses.

Reviewer #3 (Comments to the Authors (Required)):

In this manuscript, De Paola et al study the impact of the RNA binding protein Sam68 on skeletal muscle. They used a Sam68 knock out mouse model and revealed that Sam68 is essential for proper neuromuscular connections and muscle strength. The data highlight a high alteration of the neuromuscular system in Sam68^{-/-} that will be important for the field. However, in my opinion analysis-interpretations are sometimes imprecise and their completion would strengthen the manuscript.

Major points

1. The title of the manuscript is misleading "Sam68 splicing regulation insures proper motor unit establishment ...". Unless the authors make a causal link between alternative splicing dysregulations in Figure 1 and the alteration of the neuromuscular system in the rest of the paper, the title should be modified.

As suggested by the Reviewer, the title of the manuscript has been modified in "Sam68 splicing regulation contributes to motor unit establishment in the postnatal skeletal muscle".

2. Figure 2A, B: In Sam68^{-/-} mice (and to a lesser extent in WT), there are a major proportion of "NMJ"/ bungarotoxin positive element that do not colocalize with the pre-synaptic nerve terminal (synaptophysin). Could it be an indication for initiation of NMJ disruptions at 30dpp? If this is the case, the quantification in Figure 2B should distinguish the different types / degeneration stages of NMJs.

The higher density of NMJs in Sam68^{-/-} mice can be associated with the alteration in their maturation process, potentially due to defects in muscle innervation. In this regard, we observed a significant proportion of NMJ/ bungarotoxin positive element that do not colocalize with the pre-synaptic nerve terminal (synaptophysin) (Supplementary Figure 4B). This can be an indication of reduced muscle innervation, due to defects in the establishment of the junction.

3. In general, it would be important to look at the muscle fiber, the motor neuron and their junctions at earlier time points. It would give some indication in regards to the developmental stages from when Sam68^{-/-} phenotypes are expressed. It could also help in identifying the first - and perhaps the causal -element leading to the substantial muscle defects. It is of particular relevance since in their previous study (Pagliarini et al, 2015), the authors showed that ablation of Sam68 do not affect NMJs and muscle fiber sizes at 10 dpp (in non SMA mice).

We have performed motor neuron count at 8 days post partum (dpp) and found no significant difference in the Sam68ko mice, consistently with the previous report (Pagliarini et al., 2015). These data are now shown in Fig. 2G of the revised manuscript. Regarding muscle morphology, we also did not detect major changes at this stage of development, as reported in Pagliarini et al., 2015. We now discuss this issue in the revised paper (see page 14). In the preliminary phase of our study we found the first mild signs of muscle defects at 4 weeks (30 dpp), this is why we set this time as

early point. Since at 30 dpp the motor neuron reduction is already dramatic, our hypothesis is that muscle defects are subsequent to motor neuron degeneration. Given that Sam68 knockout mice are sterile and display reduced viability (Richard et al., 2005), collecting sufficient mice at multiple time points is not an easy task and would require prolonged time, which would strongly delay the revision of the current study.

4. I found sometimes difficult to make the distinction between atrophy versus lack of growth of muscle fibers (Figures 3 and 4). The authors should be more cautious with respect to their conclusions or better support them experimentally.

We agree with the Reviewer and we took advantage of his/her suggestion in the discussion of our data (see page 15), pointing out that the distinction between muscle atrophy versus lack of growth of muscle fibers can be difficult to discriminate considering only the muscle phenotype. Nevertheless, we addressed this issue by performing a series of morphological, molecular and morphometric analysis. In particular, based on our data, we can speculate that ablation of Sam68 interferes with a series of homeostatic mechanisms, leading to muscle atrophy. In particular, the reduction in muscle fibers size was associated with a denervated phenotype, as indicated by a significant proportion of NMJ/ bungarotoxin positive element that do not colocalize with the pre-synaptic nerve terminal (synaptophysin) (supplementary Figure 4B), and with a reduced number of motor neurons, which may die as a consequence of defective establishment of NMJs. This, in turn, may lead to a significant up-regulation of the master-regulatory genes involved in the promotion of muscle atrophy, thus leading to the reduced size of muscle fibers.

5. The authors mention that Sam68 is highly expressed in the motor neurons of spinal cord and revealed some substantial dysregulations of alternative splicing in Sam68^{-/-} spinal cord. Is the ablation of Sam68 in motor neurons responsible of the skeletal muscle alteration?

As indicated by the Reviewer, this is our hypothesis. The cross-talk between motor neurons and muscle is essential to accomplish proper muscular functions. We think that defects in motor neuron function/survival may underlie a significant part of the muscular phenotype (see also response to previous point). However, we cannot rule out a contribution of Sam68 function directly in the muscle. Thus, in the revised manuscript we now discuss this possibility more clearly (see page 14). Unfortunately, a conditional Sam68 knockout mouse is not available and developing one is beyond the scope of the present manuscript.

Minor points

1. Figure1A: The authors claim that Sam68 in spinal cord favors skipping of Nrnx3 exon20. However, the PCR signal appears to be saturated preventing a rigorous quantification. The authors should use an accurate quantitative method to analyze Nrnx3 exon20 alternative splicing in Sam68^{-/-}.

Indeed, Nrnx3 is not a strong target of Sam68 and we have deleted this PCR data from the manuscript. Additional targets that show more clear changes in the knockout are now shown. When primer design for qPCR was possible, we also validated these events by real time PCR.

2. Figure 2E: i) please show the whole spinal cord section for both WT and KO mice and ii) label the images.

Images of whole spinal cord sections have been added in the revised Figure 2.

3. Figure 3C, D: please add the age of the animals on the figure as well.

Animal age has been indicated in the revised Figure 3.

4. Figure 5: The authors should ensure FoxO3A and Myogenin are increased at the protein level in Sam68^{-/-} to support their conclusion.

Western blot analysis documenting Myogenin increase at the protein level was added in the revised manuscript. For FoxO3A, we did not have antibodies available.

5. Figure 7: I could not find at which age the experiments were performed.

This information has been added in the revised manuscript.

References

Chawla G, Lin CH, Han A, Shiue L, Ares M, Black DL (2009) Sam68 regulates a set of alternatively spliced exons during neurogenesis. *Mol Cell Biol* 29: 201-213

Farini D, Cesari E, Weatheritt RJ, La Sala G, Naro C, Pagliarini V, Bonvissuto D, Medici V, Guerra M, Di Pietro C et al (2020) A Dynamic Splicing Program Ensures Proper Synaptic Connections in the Developing Cerebellum. *Cell Rep* 31: 107703

Guo J, Qiu W, Soh SL, et al. Motor neuron degeneration in a mouse model of seipinopathy. *Cell Death Dis.* 2013;4(3):e535.

Ma J, Smith BP, Smith TL, Walker FO, Rosencrance EV, Koman LA (2002) Juvenile and adult rat neuromuscular junctions: density, distribution, and morphology. *Muscle Nerve* 26: 804-809

Marques MJ, Conchello JA, Lichtman JW (2000) From plaque to pretzel: fold formation and acetylcholine receptor loss at the developing neuromuscular junction. *J Neurosci* 20: 3663-3675

Pagliarini V, Pelosi L, Bustamante MB, Nobili A, Berardinelli MG, D'Amelio M, Musarò A, Sette C (2015) SAM68 is a physiological regulator of SMN2 splicing in spinal muscular atrophy. *J Cell Biol* 211: 77-90

Paronetto MP, Messina V, Barchi M, Geremia R, Richard S, Sette C (2011a) Sam68 marks the transcriptionally active stages of spermatogenesis and modulates alternative splicing in male germ cells. *Nucleic Acids Research* 39: 4961-4974

Richard S, Torabi N, Franco GV, Tremblay GA, Chen T, Vogel G, Morel M, Cl  roux P, Forget-Richard A, Komarova S et al (2005) Ablation of the Sam68 RNA binding protein protects mice from age-related bone loss. *PLoS Genet* 1: e74

Scurry AN, Heredia DJ, Feng CY, Gephart GB, Hennig GW, Gould TW (2016) Structural and Functional Abnormalities of the Neuromuscular Junction in the Trembler-J Homozygote Mouse Model of Congenital Hypomyelinating Neuropathy. *J Neuropathol Exp Neurol* 75: 334-346

Shi L, Fu AK, Ip NY (2012) Molecular mechanisms underlying maturation and maintenance of the vertebrate neuromuscular junction. *Trends Neurosci* 35: 441-453

Vercelli A, Mereuta OM, Garbossa D, et al. Human mesenchymal stem cell transplantation extends survival, improves motor performance and decreases neuroinflammation in mouse model of amyotrophic lateral sclerosis. *Neurobiol Dis.* 2008;31(3):395-405.

Witte H, Schreiner D, Scheiffele P (2018) A Sam68-dependent alternative splicing program shapes postsynaptic protein complexes. *Eur J Neurosci*

July 17, 2020

RE: Life Science Alliance Manuscript #LSA-2019-00637-TR

Prof. Maria Paola Paronetto
University of Rome Foro Italico
Department of Movement, Human and Health Sciences
Piazza Lauro de Bosis, 15
Rome, Italy 00135
Italy

Dear Dr. Paronetto,

Thank you for submitting your revised manuscript entitled "Sam68 splicing regulation contributes to motor unit establishment in the postnatal skeletal muscle". Your manuscript was re-reviewed by one of the original reviewers, and their report is attached below. We would be happy to publish your paper in Life Science Alliance pending final revisions necessary to meet our formatting guidelines.

- Please adapt the discussion to reframe the conclusions along the lines suggested by the referee - in particular regarding the alternative interpretation 'the phenotype could be caused by motorneuronal loss due to splicing of transcripts involved in neuromaintenance'.
- Please correct the sentence noted by the ref in point #1.
- Please add the control requested for S1A.
- Please respond to #2 and clarify the manuscript text where appropriate.
- Please update the figure reference in #3, if appropriate.
- Please add ORCID ID for corresponding author - you should have received instructions on how to do so
- Please add a running title and summary blurb
- Please upload your supplementary figures as separate files
- Please take another look at your callouts-there is a callout for Supp. Fig. 1C,D but there isn't a panel C,D for Fig.S1; add callouts for Figure 7D, Supp. Fig 2C-D, supp. fig. 4B; you have a callout for Table 2, but there is only a Table 1 uploaded
- Please add Tables as editable docx or excel formats
- Please add supplementary figure legends to the main manuscript text
- Please take another look at your Figures and Figure legends: Fig. 1: only panels A-F mentioned in fig. Legend, but Figure has panels A-I; Fig. 5: panel G is missing from the figure legend; Fig. S3: in legend, there is a Panel G, but this is not part of the figure
- Please add scale bars to Figure 2H, Figure S3A, and Figure S4A
- Please make sure that the origin box matches the magnification in Figure S3A

A. FINAL FILES:

B. MANUSCRIPT ORGANIZATION AND FORMATTING:

Sincerely,

Reilly Lorenz
Editorial Office Life Science Alliance
Meyerhofstr. 1
69117 Heidelberg, Germany
t +49 6221 8891 414
e contact@life-science-alliance.org
www.life-science-alliance.org

Reviewer #2 (Comments to the Authors (Required)):

Revision De Paola et al.

The authors have made an appreciable effort to meet the Reviewers requests. I feel that in general the manuscript is improved, and several questions have been satisfactorily addressed. What remains in my opinion unclear is the final message of the paper. The phenotype that they characterize could be caused by motoneuronal loss owing to dysregulated splicing of transcripts involved in neuronal maintenance, yet the focus of the manuscript seems to be on a secondary muscular atrophy. In the revised version they show additional data supporting a neurological phenotype. I think that the Discussion could have still been improved in this direction, even if the authors use a full-body KO where it is difficult to tear apart tissue-autonomous phenotypes.

A few minor issues should be corrected:

1) In the text, they write: "Spinal cord tissue was isolated from wild type and Sam68^{-/-} mice and the differential expression of Sam68 was confirmed at both RNA and protein levels (Supplementary Figure 1A and B)." However, Fig. S1B shows a western blot on muscle tissue. The sentence is misleading and should be rephrased. In addition, please add a control or the RT-PCR in S1A.

2) What is the difference between data shown in Figure 1 and S2 for Nrnx2 (Graph is also different...)? This is not clear from text or figure legends. Moreover, in the legend to Fig. 1, please clarify which transcripts have been analysed by RT- qPCR or normal RT-PCR.

3) In the sentence "In Sam68^{-/-} mice we observed a significant decrease in the overlaid signals, along with an increase in the non-overlaid signals (Supplementary Figure 2A and B) demonstrating the presence of altered junctions" referenced to the Figure is wrong. This should be Supplementary Figure S4.

Reviewer #2

Revision De Paola et al.

The authors have made an appreciable effort to meet the Reviewers requests. I feel that in general the manuscript is improved, and several questions have been satisfactorily addressed. What remains in my opinion unclear is the final message of the paper. The phenotype that they characterize could be caused by motoneuronal loss owing to dysregulated splicing of transcripts involved in neuronal maintenance, yet the focus of the manuscript seems to be on a secondary muscular atrophy. In the revised version they show additional data supporting a neurological phenotype. I think that the Discussion could have still been improved in this direction, even if the authors use a full-body KO where it is difficult to tear apart tissue-autonomous phenotypes.

We thank the Reviewer for the positive feedback on our revised manuscript and for his/her efforts in ameliorating our work. In this revised version we have now reframed our conclusions introducing a sentence explaining that the phenotype of Sam68^{-/-} mice could be caused by motor neuronal loss due to aberrant splicing of transcripts involved in neuromaintenance'.

A few minor issues should be corrected:

1) In the text, they write: "Spinal cord tissue was isolated from wild type and Sam68^{-/-} mice and the differential expression of Sam68 was confirmed at both RNA and protein levels (Supplementary Figure 1A and B)." However, Fig. S1B shows a western blot on muscle tissue. The sentence is misleading and should be rephrased. In addition, please add a control or the RT-PCR in S1A.

The sentence has been rephrased and the two tissues analyzed have been specified in the main text. Moreover, the Gapdh control has been added in the RT-PCR. We thank the Reviewer for pointing it out.

2) What is the difference between data shown in Figure 1 and S2 for Nrnx2 (Graph is also different...)? This is not clear from text or figure legends. Moreover, in the legend to Fig. 1, please clarify which transcripts have been analysed by RT-qPCR or normal RT-PCR.

Nrxn1 splicing outcome was analyzed with both RT-qPCR (Fig.1A) and RT-PCR (Fig.S2A) methods, as requested by the Reviewer, obtaining similar results. In the case of Nrnx2, we did not appreciate any change in splicing, in line with previous results from Danilenko and colleagues. In fact, Nrnx2 is a specific target of SLM2 but not Sam68 (Danilenko et al., 2017).

3) In the sentence "In Sam68^{-/-} mice we observed a significant decrease in the overlaid signals, along with an increase in the non-overlaid signals (Supplementary Figure 2A and B) demonstrating the presence of altered junctions" referenced to the Figure is wrong. This should be Supplementary Figure S4.

We thank the Reviewer for noticing this typo, that has been corrected in the revised manuscript.

July 24, 2020

RE: Life Science Alliance Manuscript #LSA-2019-00637-TRR

Prof. Maria Paola Paronetto
University of Rome Foro Italico
Department of Movement, Human and Health Sciences
Piazza Lauro de Bosis, 15
Rome, Italy 00135
Italy

Dear Dr. Paronetto,

Thank you for submitting your Research Article entitled "Sam68 splicing regulation contributes to motor unit establishment in the postnatal skeletal muscle". It is a pleasure to let you know that your manuscript is now accepted for publication in Life Science Alliance. Congratulations on this interesting work.

DISTRIBUTION OF MATERIALS:

Again, congratulations on a very nice paper. I hope you found the review process to be constructive and are pleased with how the manuscript was handled editorially. We look forward to future exciting submissions from your lab.

Sincerely,

Reilly Lorenz
Editorial Office Life Science Alliance
Meyerhofstr. 1
69117 Heidelberg, Germany
t +49 6221 8891 414
e contact@life-science-alliance.org
www.life-science-alliance.org